# Distilling Non-Autoregressive Model Knowledge for Autoregressive De Novo Peptide Sequencing

## Abstract

Autoregressive (next-token-prediction) models excel in various language generation tasks compared to non-autoregressive (parallel prediction) models. However, their advantage diminishes in certain biology-related tasks like protein modeling and de novo peptide sequencing. Notably, previous studies show that Non-Autoregressive Transformers (NAT) can largely outperform Autoregressive Transformers (AT) in amino acid sequence prediction due to their bidirectional information flow. Despite their advantages, NATs struggle with generalizing to longer sequences, scaling to larger models, and facing extreme optimization difficulties compared to AT models. Motivated by this, we propose a novel framework for directly distilling knowledge from NATs, known for encoding superior protein representations, to enhance autoregressive generation. Our approach employs joint training with a shared encoder and a specially designed cross-decoder attention module. Additionally, we introduce a new training pipeline that uses importance annealing and cross-decoder gradient blocking to facilitate effective knowledge transfer. Evaluations on a widely used 9-species benchmark show that our proposed design achieves state-of-the-art performance. Specifically, AT and NAT baseline models each excel in different types of data prediction due to their unique inductive biases. Our model combines these advantages, achieving strong performance across all data types and outperforming baselines across all evaluation metrics. This work not only advances de novo peptide sequencing but also provides valuable insights into how autoregressive generation can benefit from non-autoregressive knowledge and how next-token prediction (GPT-style) can be enhanced through bidirectional learning (BERT-style). We release our code for reproduction in the anonymous repository here: https://anonymous.4open.science/r/CrossNovo-E263.

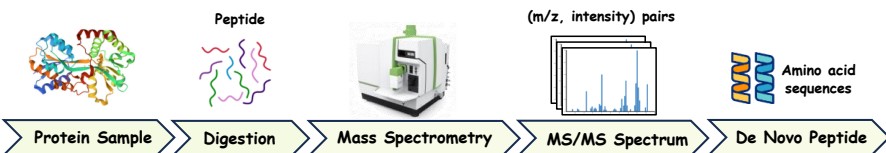

Figure 1: De novo peptide sequencing process using tandem mass spectrometry. Our goal is to predict amino acids sequence from the given spectrum as shown in the last two steps.

## 1 Introduction

Peptide (shorter protein) sequencing, entailing the inference of amino acid sequences from spectral data obtained from biological samples via tandem mass spectrometry, constitutes a fundamental element of proteomics research (Aebersold & Mann, 2003a). This process, as illustrated in Figure 1, is critically important for both foundational and applied investigations across chemistry, biology, medicine, and pharmaceutical sciences (Aebersold & Mann, 2003b; Ng et al., 2023). Traditional methods rely on database searching, where they employ a target database (Eng et al., 1994; Perkins et al., 1999; Cox & Mann, 2008; Zhang et al., 2012) to identify candidate peptide sequences matching

the query spectrum. However, these methods are constrained by the necessity of a comprehensive and accurate database, limiting their applicability in contexts such as monoclonal antibody sequence assembly (Beslic et al., 2022), novel antigen identification (Karunratanakul et al., 2019), and metaproteome sequencing where established databases are often absent for those proteins (Hettich et al., 2013). De (from) novo (scratch) peptide sequencing directly infers the peptide sequence from the spectral data, thereby circumventing the limitations associated with database-dependent algorithms.

The development of de novo peptide sequencing algorithms has continued for more than 20 years (Dančík et al., 1999). DeepNovo (Tran et al., 2017) significantly advanced de novo sequencing by integrating Convolutional Neural Networks (CNNs) and Long Short-Term Memory networks (LSTMs), showcasing the robust generalization capabilities of deep learning (LeCun et al., 2015). The introduction of Transformer models (Vaswani et al., 2017) further revolutionized the field, with Casanovo (Yilmaz et al., 2022; 2023), ContraNovo (Jin et al., 2024), and PrimeNovo (Zhang et al., 2024) employing diverse strategies to enhance performance and efficiency, achieving state-of-the-art results in de novo peptide sequencing. Building on the advances in natural language processing (NLP), previous deep learning models for peptide sequencing have predominantly leveraged autoregressive models as their generative backbone. However, autoregressive models generate sequences in a unidirectional manner, predicting the next token based on the preceding ones. This generative approach is misaligned with protein sequence formation, where each amino acid's functionality and characteristics depend on the surrounding instead of just preceding amino acids. On the other hand, Non-Autoregressive Transformer (NAT) de novo sequencing models replace causal attention with self-attention and enable bidirectional information flow and simultaneous token generation. It successfully demonstrated superior performance in the sequencing task (Zhang et al., 2024).

Despite their benefits, NAT models encounter several generative challenges in natural languages, which also extend to biological sequences. Specifically, adapting NAT models to varying sequence lengths is challenging because the prediction length is often pre-defined or pre-predicted to facilitate parallel prediction (Liu et al., 2022). Furthermore, models trained with non-autoregressive generative losses, such as Connectionist Temporal Classification (CTC) loss (Graves et al., 2006; Graves & Jaitly, 2014), do not scale as effectively as autoregressive models and suffer from frequent training failure due to their more complex optimization landscapes (Qian et al., 2020), constraining the potential to increase the model size and data volume for enhanced performance.

In this study, we introduce CrossNovo, a method designed to enhance the performance of autoregressive models in de novo sequencing by leveraging learned bidirectional latent knowledge from NAT models. To achieve this, we implement several architectural modifications. We begin by sharing the spectrum encoder between two types of decoders for teacher-student pre-training, employing techniques such as multitask learning (Ruder, 2017; Zhang & Yang, 2021) and importance annealing to foster improved learning outcomes. We then propose the first-ever cross-decoder attention module for distilling knowledge from the teacher decoder to students, with carefully designed positional encoding and gradient blocking for smooth knowledge transfer.

Our model, CrossNovo, tested on widely-used benchmarks across nine species (Tran et al., 2017; Yilmaz et al., 2022; 2023), demonstrates superior performance over both AT and NAT baselines. Combining the strengths of both model types through knowledge distillation, CrossNovo significantly improves performance on species where NAT excels but AT models struggle. Additionally, it extends AT's original advantage in sequencing species such as *human* and *mouse*. Comprehensive ablation studies validate our key contributions, while in-depth analyses showcase CrossNovo's enhanced ability to distinguish structurally similar amino acids and its generalized performance on downstream tasks such as sequencing human antibody data. These findings underscore the effectiveness of our approach and the potential of CrossNovo in advancing proteomics research.

## 2 RELATED WORK

### 2.1 AUTOREGRESSIVE AND NON-AUTOREGRESSIVE TRANSFORMERS.

The Transformer architecture (Vaswani et al., 2017) has substantially advanced the fields of sequence representation and generation. The vanilla Transformer adopts an autoregressive method for sequence generation, where each token is generated sequentially, conditioned on the preceding tokens. While the autoregressive Transformer demonstrates outstanding performance across various tasks, it suffers

from inefficiencies during inference because of its sequential generation process. To address these inefficiencies, Non-Autoregressive Transformers have been proposed (Gu et al., 2017; 2019; Ma et al., 2019; Ding et al., 2020; Gu & Kong, 2020; Huang et al., 2022; Xiao et al., 2023), which reduce inference latency by generating all tokens simultaneously, while still achieving satisfactory results.

Most existing Transformer-based de novo peptide sequencing methods utilize an autoregressive architecture (Yilmaz et al., 2022; Jin et al., 2024). A significant limitation of these methods is their vulnerability to early prediction errors of amino acids, which can lead to cumulative errors that degrade overall performance. In this study, we employ a non-autoregressive decoder to capture bidirectional information, thereby providing valuable context to the autoregressive decoder and enhancing overall performance.

## 2.2 DE NOVO PEPTIDE SQUENCING.

Early de novo sequencing methods primarily relied on graph theory and dynamic programming algorithms, incorporating various scoring functions to evaluate candidate peptide sequences (Ma et al., 2003; Frank & Pevzner, 2005; Chi et al., 2010; Ma, 2015). Although effective, these methods had limitations in handling the complexity of mass spectrometry data. DeepNovo (Tran et al., 2017) demonstrated the robust generalization capabilities of deep learning by leveraging CNNs and LSTMs to learn features from mass spectrometry data and peptide sequences. This success paved the way for numerous deep learning-based approaches in de novo peptide sequencing (Zhou et al., 2017; Yang et al., 2019; Qiao et al., 2021; Mao et al., 2023; Liu et al., 2023).

Given the notable success of Transformer (Vaswani et al., 2017) models in the deep learning field, researchers have begun to investigate their application to de novo sequencing. Transformer-based methods offer significant advantages in sequence modeling and can be broadly classified into autoregressive and non-autoregressive generation approaches. CasaNovo (Yilmaz et al., 2022; 2023) pioneered the use of Transformers for autoregressive peptide decoding. Building on this foundation, ContraNovo (Jin et al., 2024) adopted a multimodal alignment strategy to train models, achieving state-of-the-art results for autoregressive models. PrimeNovo (Zhang et al., 2024) introduced the first non-autoregressive Transformer architecture, incorporating a Precise Mass Control module to ensure precise quality control of generated peptide sequences.

Despite these advancements, adapting NAT models to varying peptide sequence lengths remains challenging due to the necessity for predefined prediction lengths to facilitate parallel computation (Liu et al., 2022). Consequently, there is an urgent need to enhance AT models to address these limitations. To this end, our research advances the field by distilling insights from non-autoregressive models to augment autoregressive models with bidirectional latent representation.

## 3 METHOD

### 3.1 PROBLEM FORMULATION AND NOTATION OVERVIEW

De novo peptide sequencing aims to generate the amino acid sequence from a given input mass spectrum (Figure 1 last 2 steps). Formally, the input set $\mathcal{S} = \{\mathcal{I}, \mathbf{c}, \mathbf{m}\}$ comprises three types of information: a set of detected mass-to-charge ratios (mz) and their corresponding intensities (g), denoted as $\mathcal{I} = \{(\text{mz}_1, \text{g}_1), (\text{mz}_2, \text{g}_2), \ldots, (\text{mz}_k, \text{g}_k)\}$; a precursor mass information, which is a float number $\mathbf{m}$; and a precursor charge information which is an integer $\mathbf{c}$. The objective is to generate the target amino acid sequence $\mathbf{A} = (\text{a}_1, \text{a}_2, \ldots, \text{a}_n)$ from the provided input $\mathcal{S}$.

### 3.2 MODEL BACKBONE

**Spectrum Encoder.** We employ a Transformer Encoder to process the input data $\mathcal{S}$. Specifically, we interpret the set of mass-to-charge ratios and intensities as a sequence of input tokens, encoding each $\text{mz}_i$ value and $\text{g}_i$ into $d$ dimensions as follows:

$$\mathbf{e}_i^0(\mathbf{v}) = \begin{cases} \sin((\mathbf{v})/(\frac{(\mathbf{v})_{\max}}{(\mathbf{v})_{\min}}(\frac{(\mathbf{v})_{\min}}{2\pi})^{\frac{2i}{d}})), & \text{for } i \leq \frac{d}{2} \\ \cos((\mathbf{v})/(\frac{(\mathbf{v})_{\max}}{(\mathbf{v})_{\min}}(\frac{(\mathbf{v})_{\min}}{2\pi})^{\frac{2i}{d}})), & \text{otherwise} \end{cases} \quad (1)$$

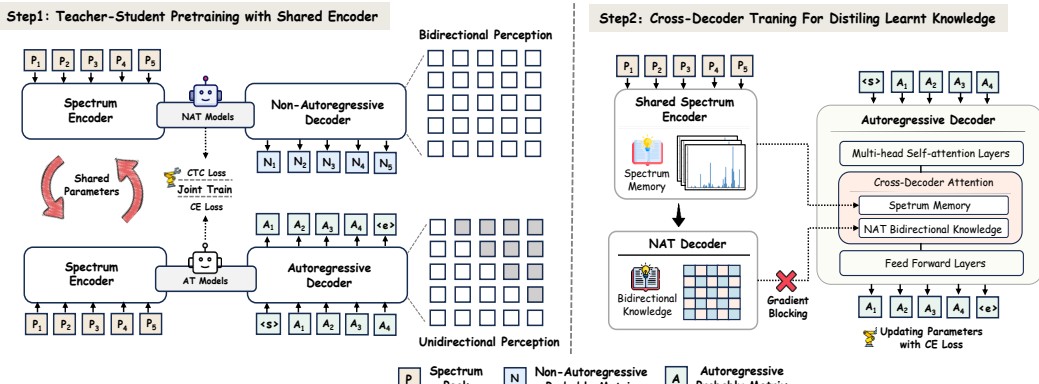

Figure 2: The architecture of CrossNovo. Step 1 involves joint training with a shared encoder in a multitask learning framework, enabling the simultaneous training of Autoregressive (AT) and Non-Autoregressive (NAT) decoders. This approach exploits the synergistic advantages of multitask learning to enhance performance. Upon convergence of both decoders, Step 2 introduces a novel knowledge distillation process, transferring insights from the NAT module to the AT module through a cross-decoder attention mechanism. Cross-decoder gradient blocking is employed throughout to optimize the training process.

where $\mathbf{v}$ is the float value to be encoded, with $\mathbf{v}_{\max}$ and $\mathbf{v}_{\min}$ being the bound value for all $\mathbf{v}$ used. The encoded mz and g at corresponding positions will be added to form the input embedding $\mathbf{E}^{(0)}$. We then subsequently feed $\mathbf{E}^{(0)}$ through $b$ self-attention layers to extract features, with each layer defined by:

$$\mathbf{E}^{(i)} = \text{SelfAttentionLayer}(\mathbf{E}^{(i-1)}) \tag{2}$$

The output embeddings from the final layer, denoted as $\mathbf{E}^{(b)} = (\mathbf{e}_1^b, \mathbf{e}_2^b, \ldots, \mathbf{e}_k^b)$, are utilized as the extracted spectrum features for the decoder.

**AT Peptide Decoder.** The autoregressive decoder is based on a standard Transformer decoder, customized for peptide prediction. Initially, the encoded amino acids $\mathbf{H}^{(0)}$ from embedding layers are processed through a causal attention block. Each layer of causal attention applies a causal mask, allowing the current position to attend only to the encoded features $\mathbf{h}_{1:t-1}$ up to that point, denoted as: $\mathbf{h}_t^{(i)} = \text{Attn}(\mathbf{h}_t^{(i-1)}, \{\mathbf{h}_{1:t-1}^{(i-1)}\})$

The newly generated features at each position, $\mathbf{h}_t^{(i)}$, then attend to the extracted spectrum features $\mathbf{E}^b$ from the encoder via a cross-attention mechanism. The output from the last decoder layer, $\mathbf{H}^{(L)}$, is subsequently used to generate the probability for each token (amino acid) as follows: $P_t(\cdot \mid \mathcal{S}, y_{1:t-1}) = \text{softmax}(\mathcal{W}\mathbf{h}_t^{(L)})$, and the current token $y_t$ will be sampled from $P_t(\cdot \mid \mathcal{S}, y_{1:t-1})$. Following the success of previous AT designs (Jin et al., 2024), we also encode the prefix mass and suffix mass information at each decoding step. This information, derived from the already decoded tokens, provides crucial biological context to the model.

**NAT Peptide Decoder.** In contrast to the autoregressive model, the NAT decoder eliminates the causal mask in the attention layer and simultaneously predicts the probability of tokens at each position. We fix the maximum generation length following previous work (Zhang et al., 2024) and input only positional embeddings at each position. The absence of true sequence embeddings as input during training is essential for transferring knowledge to the AT model, as discussed later. The embeddings $\mathbf{V}^{(0)}$ at each position undergo self-attention and cross-attention layers, akin to the process described previously. The final layer's embeddings $\mathbf{V}^{(L)}$ are then linearly projected to the vocabulary size to predict the token at each position.

### 3.3 JOINT TRAINING WITH SHARED ENCODER

Instead of training two separate models and subsequently fusing them, we adopt a joint training framework for AT and NAT. Given that the two decoders share similar learning objectives yet capture different sequence characteristics, this optimization paradigm enhances generation by leveraging

signals from both architectures. Specifically, the encoder parameters are shared between the AT and NAT decoders, and the entire network's parameters are optimized using a multitask learning loss.

**Multitask Learning Loss.** The multitask learning loss linearly combines the AT and NAT losses as follows:

$$\mathcal{L}_{\texttt{mul}} = \lambda_{\texttt{AT}}\mathcal{L}_{\texttt{AT}} + (1 - \lambda_{\texttt{AT}})\mathcal{L}_{\texttt{NAT}} \tag{3}$$

The autoregressive learning loss is based on cross-entropy:

$$\mathcal{L}_{\texttt{AT}} = -\log P(\mathbf{A} \mid \boldsymbol{\mathcal{S}}; \theta) = -\sum_{t=1}^{\mathbf{n}} \log p(\mathbf{a}_t \mid \mathbf{a}_{x<t}, \boldsymbol{\mathcal{S}}; \theta), \tag{4}$$

where $\theta$ denotes the model parameters being optimized.

For the NAT model, we utilize CTC (Graves et al., 2006) loss for capturing better intra-token connections. Specifically, we first define a max generational length $\mathbf{t}$, and the generated sequence $\mathbf{y} = (\mathbf{y}_1, \mathbf{y}_2, \cdots, \mathbf{y_t})$ is then reduced to the target length according to CTC rules. This involves merging consecutive tokens first and removing placeholder $\epsilon$ tokens. An example of such reduction on a protein sequence is $\Gamma(\text{AAT}\epsilon\text{TG}) = \text{ATTG}$. The CTC objective is to maximize probability of all $\mathbf{y}$ sequences such that $\mathbf{y}$ can be reduced to target sequence $\mathbf{A}$:

$$\mathcal{L}_{\texttt{NAT}} = -P(\mathbf{A}|\boldsymbol{\mathcal{S}}, \theta) = -\sum_{\mathbf{y}:\Gamma(\mathbf{y})=\mathbf{A}} P(\mathbf{y}|\boldsymbol{\mathcal{S}}, \theta) = \sum_{\mathbf{y}:\Gamma(\mathbf{y})=\mathbf{A}} \sum_{y_i \in \mathbf{y}} \log P(\mathbf{y}_i|\boldsymbol{\mathcal{S}}, \theta) \tag{5}$$

**Importance Annealing.** Our primary objective is to optimize for superior AT generation; thus, the NAT loss serves as an auxiliary component in the total loss calculation and should not dominate the optimization direction. In accordance with the findings of Hao et al. (2020), we employ a dynamic scheduler to adjust the weight factor $\boldsymbol{\lambda}_{\texttt{AT}}$, balancing the losses of the two decoders while prioritizing the AT network. Specifically, we increase the importance of the AT module by adjusting $\boldsymbol{\lambda}_{\texttt{AT}}$ with $\boldsymbol{\lambda}_{\texttt{AT}} = \frac{\texttt{i}}{\mathbf{T}}$. where $\texttt{i}$ represents the number of optimization iterations performed and $\mathbf{T}$ is the total number of iterations used for training the model. As a result, the weight of the NAT loss will be annealed from $1.0$ to $0.0$ over time. Initially, the NAT model dominates the optimization direction, as it effectively captures bidirectional generative information, but the AT loss will eventually take precedence as it is the target generative module.

### 3.4 Cross-Decoder Attention For Distilling Learnt Knowledge

Joint training enhances AT docoders's performance via shared gradients through the encoder during optimization. However, the learned latent knowledge of the NAT decoder does not directly influence the AT decoder's generation process. To address this shortcoming, we propose a first-ever cross-decoder attention mechanism that directly distills the learned knowledge of the NAT into the AT module, once joint training has converged, we novelly replace the cross-attention layer in AT decoder with the following:

$$\mathbf{h}_t^{\text{update}} = \text{CrossAttn}(\mathbf{h}_t^{(i)}, \{\mathbf{E}_{p\{41:41+k\}}^{(b)} \oplus \mathbf{V}_{p\{1:40\}}^{(L)}\}) \tag{6}$$

where $\oplus$ denotes the concatenation operation over the sequence length dimension. Given that the NAT decoder operates with a fixed generation length of 40, as per the settings of previous NAT (Zhang et al., 2024) based sequencing model, NAT decoder latent knowledge $\mathbf{V}^{(L)}$ comprises 40 embedding vectors. We apply a sinusoidal positional encoding to these vectors, ranging from positions 1 to 40, to obtain positional informed NAT knowledge $\mathbf{V}_{p\{1:40\}}^{(L)}$. Similarly, we encode the spectrum features with positional information to get $\mathbf{E}_{p\{41:41+k\}]}^{(b)}$ with $k$ being the length of the spectrum. This positional encoding separates the information from NAT decoder and Spectrum Encoder, thereby allowing the AT decoder to leverage it separately for better information usage.

In this distillation phase, our optimization objective is solely AT loss $\mathcal{L}_{\texttt{AT}}$, since the NAT decoder has already been fully optimized during the joint training.

**Gradient Blocking.** In our investigation of integrating information flow from the NAT to the AT via an attention mechanism, we observed a marked degradation in the performance of the NAT decoder using solely the AT loss. This deterioration adversely affected the overall network performance.

We hypothesize that this degradation occurs because the gradient from the AT's cross-entropy loss directly influences the optimization trajectory of the NAT decoder through the connected attention mechanism. This optimization objective conflicts with the NAT's original CTC objective. To mitigate this issue, we block the gradient backpropagation $\mathbf{V}^{(L)}$ as:

$$h_t^{\text{update}} = \text{CrossAttn}(h_t^{(i)}, \{\mathbf{E}_{p\{41:41+k\}}^{(b)} \oplus \mathbb{GB}(\mathbf{V}_{p\{1:40\}}^{(L)})\}) \tag{7}$$

where $\mathbb{GB}$ stands for gradient blocking through computation-graph detaching operations. This approach allows the AT decoder to leverage the learned information from $\mathbf{V}^{(L)}$ without modifying the construction of $\mathbf{V}^{(L)}$. With this modification, we observed smoother loss convergence and improved performance.

### 3.5 CAN WE REVERSE THE DIRECTION OF KNOWLEDGE DISTILLATION

An intriguing question arises: can we distill the knowledge from the AT into the NAT to further enhance the NAT decoder's performance? While it is feasible to modify the attention layer in the NAT decoder for this purpose, potential information leakage must be considered. During AT training, the true label is fed into the AT decoder. Although this does not leak information for the AT model itself due to the causal mask, performing cross-attention will inevitably reveal the true token information to the NAT. Conversely, since the NAT model does not take any true sequence as input, attending to its latent representation does not risk leaking the true target sequence during distillation.

## 4 EXPERIMENTS

### 4.1 DATASETS

To train our model in a fair comparison setting with baseline models, we followed previous work (Yilmaz et al., 2023; Zhang et al., 2024) and utilized the same training set, MassIVE-KB dataset (Wang et al., 2018), for training our model. This dataset was selected owing to its substantial repository of 30 million high-resolution peptide-spectrum matches (PSMs) obtained from various instruments. To validate our model and benchmark it against leading de novo peptide sequencing methodologies, we employed two widely accepted (Yilmaz et al., 2022; 2023; Zhang et al., 2024; Zhou et al., 2017) benchmark datasets: 9-species-v1 and 9-species-v2 revised. The 9-species-v1 dataset comprises approximately 1.5 million mass spectra derived from nine distinct experiments. The 9-species-v2 dataset significantly enhances the number and quality of spectra in the test data compared to the original 9species, encompassing a broader spectrum of data distributions with a more rigorous data annotation process.

### 4.2 EXPERIMENTS SETUP

**Implementation Details.** In this study, all inputs, including peaks and amino acids, are embedded into a 400-dimensional space. The spectrum encoder, NAT decoder, and AT decoder each consist of nine layers of Transformer architecture, utilizing an eight-head multi-head attention mechanism with hidden dimensions of 1024. Training was conducted on eight NVIDIA A100 80GB GPUs with an initial learning rate of 5e-4. To ensure training stability, a linear warm-up phase followed by a cosine decay schedule was applied. Model parameters were optimized using the AdamW optimizer (Kingma & Ba, 2014).

**Evaluation Metrics.** To evaluate the accuracy of our model's predictions, we followed *all previous work* and employed a suite of metrics at both the amino acid (AA) and peptide levels. At the AA level, correctly predicted amino acids, denoted as $\mathbf{M}_{\text{AA}}$, were identified. An amino acid was considered correctly predicted if its mass deviation from the actual amino acid was less than 0.1 Da, and any prefix or suffix mass difference did not exceed 0.5 Da relative to the corresponding segment of the actual peptide sequence. The accuracy was then calculated using $\frac{\mathbf{M}_{\text{AA}}}{\mathbf{T}_{\text{AA}}}$, where $\mathbf{T}_{\text{AA}}$ is the total number of predicted amino acids. At the peptide level, a peptide was considered accurately predicted if all its constituent amino acids matched their true counterparts. Here, $\mathbf{M}_{\text{pep}}$ denotes the number of peptides with all amino acids correctly matched in a given dataset. Peptide precision was defined as $\frac{\mathbf{M}_{\text{pep}}}{\mathbf{T}_{\text{pep}}}$, where $\mathbf{T}_{\text{pep}}$ represents the total number of peptides in the dataset. This metric provided a comprehensive evaluation of the model's performance at the peptide level.

| Metrics | Architect | Methods | *Mouse* | *Human* | *Yeast* | *M.mazei* | *Honeybee* | *Tomato* | *Rice bean* | *Bacillus* | *C. bacteria* | **Average** |
|---|---|---|---|---|---|---|---|---|---|---|---|---|
| **Amino Acid Precision** | DB | Peaks | 0.600 | 0.639 | 0.748 | 0.673 | 0.633 | 0.728 | 0.644 | 0.719 | 0.586 | 0.663 |
| | NAR | Prime. | 0.784 | 0.729 | 0.802 | 0.801 | 0.763 | 0.815 | 0.822 | 0.846 | 0.734 | 0.788 |
| | AR | Deep. | 0.623 | 0.610 | 0.750 | 0.694 | 0.630 | 0.731 | 0.679 | 0.742 | 0.602 | 0.673 |
| | | Point. | 0.626 | 0.606 | 0.779 | 0.712 | 0.644 | 0.733 | 0.730 | 0.768 | 0.589 | 0.687 |
| | | Casa. | 0.689 | 0.586 | 0.684 | 0.679 | 0.629 | 0.721 | 0.668 | 0.749 | 0.603 | 0.667 |
| | | Casa.V2 | 0.760 | 0.676 | 0.752 | 0.755 | 0.706 | 0.785 | 0.748 | 0.790 | 0.681 | 0.739 |
| | | Contra. | 0.798 | 0.771 | 0.797 | 0.799 | 0.745 | 0.810 | 0.807 | 0.828 | 0.711 | 0.785 |
| | | **Ours** | **0.816** | **0.800** | **0.814** | **0.826** | **0.785** | **0.830** | **0.831** | **0.856** | **0.740** | **0.811** |
| **Peptide Recall** | DB | Peaks | 0.197 | 0.277 | 0.428 | 0.356 | 0.287 | 0.403 | 0.362 | 0.387 | 0.203 | 0.322 |
| | NAR | Prime. | 0.567 | 0.574 | 0.697 | 0.650 | 0.603 | **0.697** | 0.702 | 0.721 | **0.531** | 0.638 |
| | AR | Deep | 0.286 | 0.293 | 0.462 | 0.422 | 0.330 | 0.454 | 0.436 | 0.449 | 0.253 | 0.376 |
| | | Point. | 0.355 | 0.351 | 0.534 | 0.478 | 0.396 | 0.513 | 0.511 | 0.518 | 0.298 | 0.439 |
| | | Casa. | 0.426 | 0.341 | 0.490 | 0.478 | 0.406 | 0.521 | 0.506 | 0.537 | 0.330 | 0.448 |
| | | Casa.V2 | 0.483 | 0.446 | 0.599 | 0.557 | 0.493 | 0.618 | 0.589 | 0.622 | 0.446 | 0.539 |
| | | Contra. | 0.567 | 0.622 | 0.674 | 0.630 | 0.576 | 0.672 | 0.677 | 0.688 | 0.486 | 0.621 |
| | | **Ours** | **0.596** | **0.661** | **0.698** | **0.660** | **0.610** | 0.695 | **0.716** | **0.726** | 0.518 | **0.654** |

Table 1: Comparison of the performance of CrossNovo and baseline methods on 9-species-v1 test set. The bold font indicates the best performance.

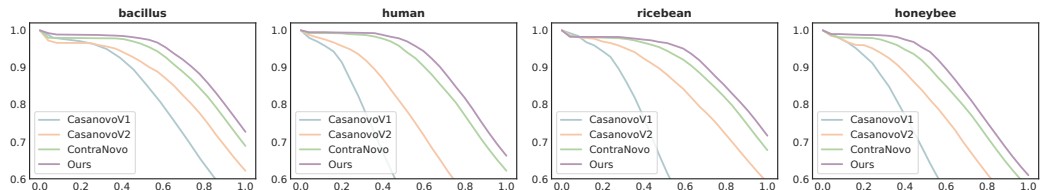

Figure 3: Peptide Precision-Coverage Curves. Due to space constraints, the full plots are provided in the Appendix under the section titled "Peptide Precision-Coverage Curves". Across all subplots, the green lines are consistently positioned above the blue and orange lines, illustrating the superior performance of our model in peptide recall over coverage levels presented by other models.

**Baselines.** Our rigorous evaluation compared our model against a wide range of approaches, which are categorized into three main types: Database (DB), Autoregressive Generation (AR), and Non-Autoregressive Generation (NAR) methods. The DB approach, exemplified by Peaks (Ma et al., 2003), utilizes tandem mass spectra from tryptic peptides for protein database searches. AR methods have made significant advancements in recent years. DeepNovo (Tran et al., 2017) was a pioneer in using CNN and LSTM architectures in this domain. Subsequent models built upon this foundation: Point-Novo (Qiao et al., 2021) improved the handling of varying resolution levels, while Casanovo (Yilmaz et al., 2022) and its enhanced version CasanovoV2 (Yilmaz et al., 2023) introduced transformer-based architectures. CasanovoV2 further incorporated beam search for improved accuracy. The most recent AR model, ContraNovo (Jin et al., 2024), introduced contrastive learning and additional amino acid mass information, pushing the boundaries of AR performance. In contrast to AR methods, the NAR approach is a recent innovation in the field. PrimeNovo (Zhang et al., 2024), the pioneering NAR method, achieves state-of-the-art precision in peptide sequence generation.

## 4.3 RESULTS

**Performance on 9-Species-v1 Benchmark Dataset.** As seen in Table 1, our trained model, Cross-Novo, demonstrated superior performance at both the amino acid level and the peptide level, achieving new state-of-the-art results on this dataset.

Specifically, our model significantly outperforms all previous autoregressive models across all species and evaluation metrics, establishing it as the leading autoregressive sequencing tool. With an average amino acid recall improved from 0.785 to 0.811 and peptide recall improved from 0.621 to 0.654, our method proves to deliver better autoregressive generational performance. Additionally, the recall-coverage graph in Figure 6 further demonstrates the superiority of our model, showing dominant performance over all previous models at all coverage levels.

Moreover, our knowledge distillation techniques successfully bridge the gap between AT and NAT. Our model not only exceeds NAT in amino acid precision across all species but also surpasses NAT in peptide recall for all but two species, where it remains highly competitive. Furthermore, we observe that our combined training and cross-decoder module have granted CrossNovo the advantages of both AT and NAT in predicting peptides of different species. Specifically, in Human and Mouse, where AT models performed significantly better than NAT models, CrossNovo extends this advantage by further outperforming NAT by 9%. In other species where NAT models performed better than AT models, CrossNovo leverages NAT strengths to increase its prediction accuracy by 1-3%.

Overall, our model's ability to integrate the strengths of both AR and NAR paradigms makes it a robust and adaptable solution. This dual capability ensures its effectiveness across diverse species, making it a valuable tool for wide-ranging biological applications.

**Performance on 9-Species-v2 Benchmark Dataset.** Our study evaluates the performance of various models on the 9-Species-v2 benchmark dataset, which features a richer diversity of modified amino acids and superior quality compared to the 9-Species-v1 benchmark dataset. The results across nine species are presented in Table 2. Our proposed model achieves an average precision of 0.906, the highest among the models tested. In terms of peptide sequence recall, our model also demonstrates significant improvements over the baseline models, achieving an average recall of 0.786. These results indicate that our proposed autoregressive model with bidirectional sequence modeling capabilities significantly outperforms the baseline models in both precision and recall metrics across all species in the 9-Species-v2 dataset.

| Metrics | Architect | Methods | Mouse | Human | Yeast | M.mazei | Honeybee | Tomato | Rice bean | Bacillus | C.bacteria | Average |
|---|---|---|---|---|---|---|---|---|---|---|---|---|
| **Amino Acid Precision** | NAR | Prime. | 0.839 | 0.893 | 0.932 | 0.908 | 0.862 | 0.909 | 0.931 | 0.921 | 0.827 | 0.891 |
| | AR | Casa.V2 | 0.813 | 0.872 | 0.915 | 0.877 | 0.823 | 0.891 | 0.891 | 0.888 | 0.791 | 0.862 |
| | | Contra. | 0.839 | 0.920 | 0.919 | 0.896 | 0.848 | 0.898 | 0.913 | 0.901 | 0.807 | 0.882 |
| | | **Ours** | **0.857** | **0.937** | **0.939** | **0.920** | **0.880** | **0.914** | **0.939** | **0.927** | **0.837** | **0.906** |
| **Peptide Recall** | NAR | Prime. | 0.627 | 0.795 | 0.884 | 0.812 | 0.742 | **0.824** | 0.837 | 0.849 | **0.626** | 0.777 |
| | AR | Casa.V2 | 0.555 | 0.712 | 0.837 | 0.754 | 0.669 | 0.783 | 0.772 | 0.793 | 0.558 | 0.714 |
| | | Contra. | 0.616 | 0.820 | 0.854 | 0.780 | 0.711 | 0.794 | 0.799 | 0.815 | 0.575 | 0.752 |
| | | **Ours** | **0.651** | **0.850** | **0.885** | **0.819** | **0.751** | 0.816 | **0.847** | **0.850** | 0.607 | **0.786** |

Table 2: Comparison of the performance of CrossNovo and baseline methods on 9-species-v2 test set. The bold font indicates the best performance.

**Performance of Amino Acids with Similar Masses.** Accurately differentiating between amino acids with very similar masses is crucial for achieving precise outcomes in peptide sequencing. We conducted rigorous evaluations to assess the performance of CrossNovo on correctly predicting these amino acids. Specifically, our objective was to determine the efficacy of CrossNovo in distinguishing these challenging cases. For a comprehensive comparison, we also evaluated the performance of all AT models. The results, depicted in Figure 4, demonstrate CrossNovo's exceptional performance. The con-

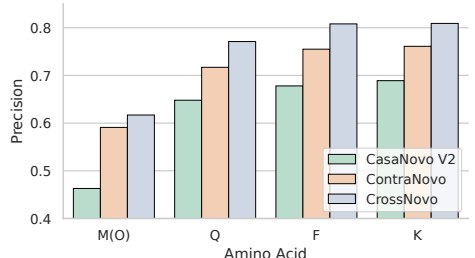

Figure 4: The precision comparison of CrossNovo with all AT models on amino acids with similar masses.

sistent better performance in identifying these easily mistaken amino acids further showcase the accuracy of CrossNovo in a more fine-grained level.

**Sensitivity to Beam Size.** We investigated the impact of varying beam sizes on the accuracy of de novo peptide sequencing using CrossNovo across the 9-species-v1 benchmark. Table 5 demonstrates that increasing beam size enhances both Amino Acid Precision and Peptide Recall. However, the advan-

| Metric | Beam Size | | | | | |
|---|---|---|---|---|---|---|
| | 1 | 3 | 5 | 7 | 9 | 11 |
| AA Precision | 0.784 | 0.804 | **0.811** | 0.810 | 0.810 | **0.811** |
| Peptide Recall | 0.634 | 0.651 | **0.654** | **0.654** | **0.654** | 0.653 |

Table 3: Effect of different beam sizes on CrossNovo.

tages of excessively large beam sizes are limited and may even lead to slight decreases in recall, potentially due to the Seq2Seq model's exposure bias (Ranzato et al., 2015; Zhang et al., 2019; Meister et al., 2020). Considering the trade-off between prediction performance and computational efficiency, we determine that a beam size of 5 is optimal, as it achieves high prediction accuracy while maintaining lower computational costs compared to larger beam sizes. For a detailed results, please refer to the "Influence of Various Beam Sizes" section in the Appendix.

**Ablation Study.** The ablation study in Table 4 evaluates the effects of different proposed modules on performance. CrossNovo achieves its highest precision scores when both the cross decoder with gradient blocking and the shared encoder are utilized. In contrast, omitting the shared encoder while retaining the cross decoder and gradient blocking significantly reduces precision. Additionally, the absence of gradient blocking leads to training failure due to gradi-

| Gradient Blocking | Cross Decoder | Shared Encoder | Amino acid Precision | Peptide Precision |
|---|---|---|---|---|
| | | ✓ | 0.795 | 0.643 |
| | ✓ | ✓ | ✗ | ✗ |
| ✓ | ✓ | | 0.698 | 0.546 |
| ✓ | ✓ | ✓ | **0.811** | **0.654** |

Table 4: Results of the ablation study showing the effects of different model configurations on amino acid and peptide precision. The '✗' indicates training failure due to gradient explosion.

ent explosion, as indicated by '✗'. These findings underscore the essential role of the cross-decoder with gradient blocking and the shared encoder in stabilizing and enhancing model performance.

**Downstream Task.** To demonstrate the generalizibity of proposed algorithm and its applicalibity, We further apply CrossNovo to a downstream task of identifying peptides in animal antibody data Beslic et al. (2023). Obtaining the sequence information of antibodies is crucial for understanding the structural basis of antibody-antigen binding, recognition, and interaction Chiu et al. (2019). However, existing methods for antibody protein sequencing rely on mRNA extraction from hybridoma cells Peng et al. (2021), which can be challenging. De novo peptide sequencing offers a faster and more accurate alternative by predicting peptides using Tandem Mass Spectrometry.

We utilize a publicly available human antibody dataset Tran et al. (2016), which includes the light chain (LC) and heavy chain (HC) antibody proteins, each digested into peptides using various enzymes. We apply CrossNovo and several state-of-the-art baseline models to evaluate this dataset. None of these models were trained on antibody data, and all perform purely zero-shot inference. As shown in Figure

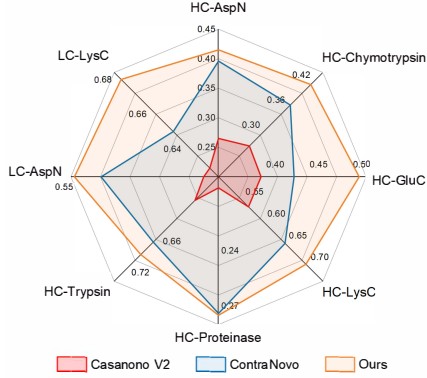

Figure 5: The peptide recall comparison of all models on Human antibody data. HC stands for heavy chain antibody and LC stands for light chain antibody. Each axis represents a different type of enzyme digestions for these antibody protein.

5, CrossNovo significantly outperforms the baseline models in human antibody sequencing, achieving up to a 5% improvement in both peptide recall and AA precision (Appendix Table 7). We also analyzed performance on mouse antibodies, with results detailed in the Appendix.

## 5 CONCLUSION

In conclusion, our research presents CrossNovo, a novel approach that significantly advances de novo peptide sequencing. By effectively integrating bidirectional latent knowledge from NAT to AT, CrossNovo leverages the strengths of both AT and NAT models. Through innovative architectural modifications, including joint training with a shared encoder and a newly designed cross-decoder attention module, CrossNovo demonstrates superior performance across diverse species, surpassing both AT and NAT baselines. Comprehensive ablation studies confirm the efficacy of our approach, while detailed analyses highlight CrossNovo's ability to accurately discriminate between similar amino acids. These contributions underscore the model's potential to drive future innovations in proteomics research, offering a powerful tool for foundational and applied investigations.

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

# A APPENDIX

## A.1 INFLUENCE OF VARIOUS BEAM SIZES

Based on the experimental results presented in Table 5, increasing the beam size generally enhances both Amino Acid Precision and Peptide Recall across various species. At both the amino acid and peptide levels, accuracy tends to improve with larger beam sizes, though it stabilizes later, exhibiting only minor increments or remaining unchanged.

For Amino Acid Precision, increasing the beam size from 1 to 3 significantly boosts average precision from 0.784 to 0.804, a rise of 0.020. Further increases up to a beam size of 11 yield smaller gains, with the highest precision of 0.811 observed at beam sizes of 5 and 11, but the rate of improvement diminishes. In terms of Peptide Recall, increasing the beam size from 1 to 3 raises average recall from 0.634 to 0.651, an increase of 0.017. Beyond a beam size of 3, improvements are marginal, with the highest recall of 0.654 achieved at beam sizes of 5, 7, and 9. Some species exhibit slight recall decreases at larger beam sizes, likely due to the Seq2Seq model's exposure bias Ranzato et al. (2015); Zhang et al. (2019); Meister et al. (2020). The model, trained with Teacher Forcing, consistently receives the correct prior output during training but must generate its own during inference, leading to potential deviations from the optimal solution. Larger beam sizes can mitigate this issue, but excessively large sizes might cause overfitting and hinder generalization.

While larger beam sizes can enhance prediction performance, they also increase inference costs. To balance effectiveness and speed, we selected a beam size of 5 for our experiments. With this size, the model achieves high performance metrics, with an average precision of 0.811 and an average recall of 0.654, showing minimal differences compared to larger beam sizes. Additionally, compared to beam sizes of 9 or 11, a beam size of 5 offers faster inference speed and lower computational demands, maintaining prediction performance while optimizing efficiency. Therefore, considering both performance and computational costs, a beam size of 5 is considered optimal, achieving an effective balance between accuracy and efficiency.

| Metrics | Species | 1-Beam | 3-Beam | 5-Beam | 7-Beam | 9-Beam | 11-Beam |
|---|---|---|---|---|---|---|---|
| | Bacillus | 0.829 | 0.850 | 0.856 | 0.854 | 0.854 | 0.855 |
| | Clambacteria | 0.713 | 0.728 | 0.740 | 0.734 | 0.734 | 0.734 |
| | Honeybee | 0.758 | 0.779 | 0.785 | 0.785 | 0.785 | 0.786 |
| | Human | 0.766 | 0.792 | 0.800 | 0.800 | 0.802 | 0.802 |
| Amino | M.mazei | 0.801 | 0.819 | 0.826 | 0.824 | 0.825 | 0.824 |
| Acid | Mouse | 0.794 | 0.813 | 0.816 | 0.816 | 0.816 | 0.817 |
| Precision | Ricebean | 0.793 | 0.820 | 0.831 | 0.827 | 0.828 | 0.828 |
| | Tomato | 0.812 | 0.826 | 0.830 | 0.830 | 0.832 | 0.832 |
| | Yeast | 0.791 | 0.809 | 0.814 | 0.815 | 0.815 | 0.816 |
| | **Average** | 0.784 | 0.804 | **0.811** | 0.810 | 0.810 | **0.811** |
| | Bacillus | 0.706 | 0.725 | 0.726 | 0.727 | 0.726 | 0.726 |
| | Clambacteria | 0.502 | 0.517 | 0.518 | 0.519 | 0.519 | 0.518 |
| | Honeybee | 0.591 | 0.607 | 0.610 | 0.610 | 0.610 | 0.610 |
| | Human | 0.632 | 0.657 | 0.661 | 0.663 | 0.664 | 0.663 |
| Peptide | M.mazei | 0.642 | 0.658 | 0.660 | 0.660 | 0.660 | 0.660 |
| Recall | Mouse | 0.577 | 0.593 | 0.596 | 0.596 | 0.596 | 0.595 |
| | Ricebean | 0.686 | 0.712 | 0.716 | 0.717 | 0.717 | 0.717 |
| | Tomato | 0.682 | 0.694 | 0.695 | 0.694 | 0.694 | 0.694 |
| | Yeast | 0.684 | 0.696 | 0.698 | 0.698 | 0.698 | 0.698 |
| | **Average** | 0.634 | 0.651 | **0.654** | **0.654** | **0.654** | 0.653 |

Table 5: Comparison of Amino Acid Precision and Peptide Recall for 9-Species-v1 at Various Beam Sizes

## A.2 PRECISION-COVERAGE CURVES

To evaluate the efficacy of our model, we utilize Precision-Coverage curves, which offer insights into performance across various species. A full visual representation of CrossNovo's outstanding

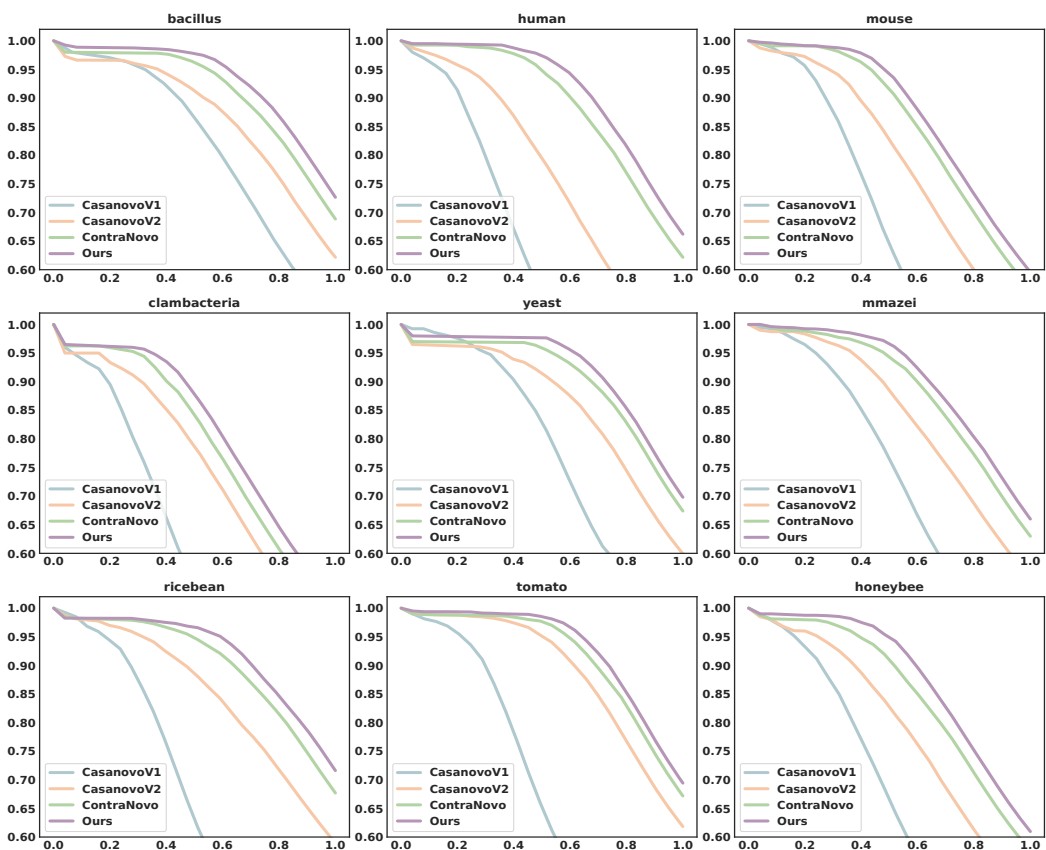

Figure 6: Peptide Precision-Coverage Curves for Various Species. Across all subplots, the green lines are consistently positioned above the blue and orange lines, illustrating the superior performance of our model in peptide recall over varying coverage levels.

performance is depicted in the Precision-Coverage curve shown in Figure 6. The horizontal axis represents coverage, while the vertical axis represents peptide recall. The blue line indicates the performance of Casanovo V2, the orange line represents ContraNovo, and the green line shows the performance of our model. Across all subplots, the green lines are consistently positioned above the blue and orange lines, illustrating the superior performance of our model in peptide recall over varying coverage levels. This consistent outperformance suggests potential for more accurate peptide identification, which could enhance biological insights.

## A.3 DOWNSTREAM TASKS

**Data.** The Human IgG1 antibody dataset (IgG1-Human) consists of mass spectrometry data collected using the LTQ Orbitrap instrument. Ionization was performed through higher-energy collisional dissociation (HCD), and the resulting peptide fragments were captured at a resolution of 17,500 FWHM. The dataset features peptides digested by a variety of proteolytic enzymes, including trypsin, chymotrypsin, asp-N, lys-C, glu-C, and proteinase K. We perform the evaluation on donwloaded data with no processing.

The Mouse IgG1 antibody dataset (WIgG1-Mouse) was similarly analyzed using an LTQ Orbitrap mass spectrometer with HCD ionization and the same resolution of 17,500 FWHM. In this dataset, the mouse peptides were digested with trypsin, asp-N, and chymotrypsin to generate a comprehensive proteomic profile. We also used downloaded data for evaluation of all tested models.

**Results.** As shown in Tables 7 and 6, CrossNovo consistently achieves superior performance across both amino acid-level precision and peptide-level recall when compared to the baseline methods.

| Metrics | Methods | HC | | | LC | Average |
| | | AspN | Chymotrypsin | Trypsin | AspN | |
|---|---|---|---|---|---|---|
| Amino Acid Precision | Casa.V2 | 0.714 | 0.591 | 0.723 | 0.668 | 0.674 |
| | Contra. | 0.750 | 0.612 | 0.650 | 0.649 | 0.665 |
| | **Ours** | **0.769** | **0.640** | **0.747** | **0.724** | **0.720** |
| Peptide Recall | Casa.V2 | 0.557 | 0.483 | 0.636 | 0.456 | 0.533 |
| | Contra. | 0.649 | 0.545 | 0.671 | 0.519 | 0.596 |
| | **Ours** | **0.662** | **0.577** | **0.699** | **0.581** | **0.630** |

Table 6: Comparison of the performance of CrossNovo and baseline methods on WIgG1-Mouse. The bold font indicates the best performance.

| Metrics | Methods | HC | | | | | | LC | | Average |
| | | AspN | Chymotrypsin | GluC | LysC | Proteinase | Trypsin | AspN | LysC | |
|---|---|---|---|---|---|---|---|---|---|---|
| Amino Acid Precision | Casa.V2 | 0.520 | 0.472 | 0.605 | 0.757 | 0.354 | 0.759 | 0.666 | 0.778 | 0.642 |
| | Contra. | 0.580 | 0.565 | 0.642 | 0.790 | 0.348 | 0.787 | 0.702 | 0.793 | 0.676 |
| | **Ours** | **0.613** | **0.617** | **0.694** | **0.814** | **0.367** | **0.803** | **0.719** | **0.807** | **0.702** |
| Peptide Recall | Casa.V2 | 0.265 | 0.274 | 0.399 | 0.569 | 0.206 | 0.595 | 0.325 | 0.625 | 0.446 |
| | Contra. | 0.396 | 0.372 | 0.437 | 0.653 | 0.274 | 0.675 | 0.499 | 0.646 | 0.529 |
| | **Ours** | **0.415** | **0.421** | **0.512** | **0.701** | **0.275** | **0.699** | **0.544** | **0.676** | **0.560** |

Table 7: Comparison of the performance of CrossNovo and baseline methods on IgG1-Human. The bold font indicates the best performance.

Specifically, for the Mouse dataset, CrossNovo demonstrates a notable improvement in peptide recall, achieving up to a 6% increase for the AspN enzyme on the light chain (LC) protein. Similarly, in the Human dataset, the peptide recall improvement is up to 4.5% for the AspN enzyme on the LC protein.

The performance gain is particularly pronounced for light chain proteins across both species, with Cross-Novo showing higher overall precision and recall. The average amino acid precision for the Mouse dataset reaches 0.720, while the peptide recall is boosted to 0.630. For the Human dataset, CrossNovo attains an average precision of 0.702, with a peptide recall of 0.560, further underscoring its effectiveness over baseline approaches.

These differences in performance are even more apparent in Figure 7, where CrossNovo's enhancements, particularly on the light chain proteins, can be clearly visualized. The consistency in performance improvements across both datasets highlights CrossNovo's ability to handle diverse proteolytic enzymes with high accuracy, especially in cases involving the light chains.

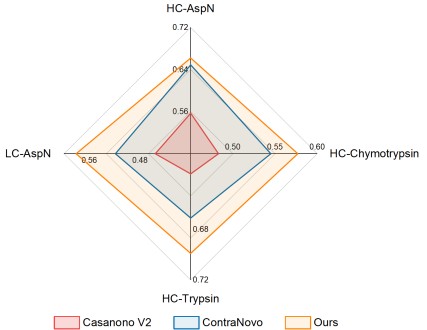

Figure 7: The comparison of performance of models in mouse antibody data.

