# OpenReview forum: "Distilling Non-Autoregressive Model Knowledge for Autoregressive De Novo Peptide Sequencing"
_ICLR.cc/2025/Conference — Submitted to ICLR 2025_

### Official Review · Reviewer_irZV · 2024-10-28

**Soundness:** 4
**Presentation:** 4
**Contribution:** 3
**Rating:** 6
**Confidence:** 4

**Summary:**

This paper proposes CrossNovo, a novel framework for distilling the bidirectional information of non-autoregressive transformers (NAT) for autoregressive transformers (AT). The training scheme consists of two parts: (1) a pre-training scheme for spectrum encoder + AT with regular cross-entropy loss and spectrum encoder + NAT with CTC loss with shared encoder parameters, and (2) cross-attention that transfers the final layer NAT embeddings to AT while freezing the NAT decoder. Experiments show SOTA results across all metrics.

**Strengths:**

- The writing is very clear, with neat figures, making the paper easy to follow.
- The motivation for the proposed method is well-explained.
- The results are strong, with thorough evaluations.

**Weaknesses:**

- The two-step training scheme, while effective, is more time-consuming than previous methods.
- The proposed method is a straightforward and expected combination of existing techniques (e.g., knowledge distillation is essentially a cross-attention with additional NAT embeddings).

Minor (typos):
- Figure 2: “Step 2: Cross-Decoder *Traning* for *Distiling* Learned Knowledge”, "*Probably* Matrix"
- Figure 3 caption: Shouldn’t it be purple lines instead of green lines?

**Questions:**

How much extra training time does CrossNovo need compared to existing end-to-end methods?

---

> ### Author Response · Authors · 2024-11-18
> **Reply to reviewer irZV**
>
> Thank the reviewer for time and effort in reading our work, we provide detailed responses below to your raised questions and concerns:
>
> > The two-step training scheme, while effective, is more time-consuming than previous methods.
>
>   Thank you for raising this point. We will include a discussion on the additional computational costs for both training and inference in our revised manuscript. Indeed, as our method involves a two-stage training process and requires two decoders to be trained, it is more time-consuming compared to previous methods.
>
>   Specifically, training the AT decoder alone requires 130 epochs, with 1 million data samples per epoch, while the NAT decoder alone requires 150 epochs. When jointly trained, the total number of epochs is reduced to 120, though each epoch takes approximately 1.3 times longer than training a single decoder. The second phase, which involves knowledge distillation, requires an additional 20 epochs. Altogether, the total training time is approximately 30% greater than training a single AT model when using the same setup of 8 A100 GPUs.
>
>   However, this modest increase in training time yields significant benefits, with an improvement of over 3% in peptide-level accuracy—a substantial gain in this field, where accuracy improvements are notoriously challenging to achieve.
>
>   Regarding inference, our approach incurs minimal additional computational cost. This is because inference does not involve backpropagation, and the NAT decoder operates efficiently during inference, as it does not decode tokens sequentially but instead computes features in a single forward pass. Consequently, the time increase during inference, once the model is fully trained, is almost negligible. This is an important consideration for biological research, where computational efficiency during deployment is critical.
> We appreciate this constructive feedback and will ensure these points are clearly addressed in our manuscript. Thank you for the opportunity to clarify this aspect of our work.
>
>
> > The proposed method is a straightforward and expected combination of existing techniques (e.g., knowledge distillation is essentially a cross-attention with additional NAT embeddings).
>
>   Thank you for this comment. The intention behind proposing this framework was to introduce a clear and effective approach for efficiently improving this critical task in protein research. In protein-based biological research, usability and performance are of paramount importance, often outweighing other considerations. Tools developed for this field are heavily utilized by biologists directly in laboratory analyses and protein discovery workflows. For example, the baseline tool DeepNovo, which employs a relatively simple LSTM architecture, has seen extensive application in such tasks since its development.
>
>
>   Our method demonstrates a significant improvement of approximately 3% in whole-sequence-level accuracy, as shown in benchmark progression. This improvement represents a substantial advance, pushing accuracy to new heights. Such a performance gain is noteworthy in this domain, where even incremental improvements have a major impact. Moreover, our open-sourced software ensures that the benefits of our work are immediately accessible and applicable to the broader biological research community.
>
>
>   We believe that the clarity of our method and its straightforward design are among its strengths. By avoiding overly complex or redundant modules, we have ensured that our approach remains practical and efficient for downstream tasks. This simplicity not only makes our method more user-friendly for biologists but also lays a solid foundation for future developers to build upon. Thank you again for raising this point, and we hope this explanation helps clarify the motivation and value of our approach.
>
>
> > Minor (typos): Figure 2: “Step 2: Cross-Decoder Traning for Distiling Learned Knowledge”, "Probably Matrix"  Figure 3 caption: Shouldn’t it be purple lines instead of green lines?
>
> We sincerely thank the reviewer for their careful reading and attention to detail! We have noted these typos and will correct them in our revised manuscript. Thank you again for bringing them to our attention.
>
>
> > How much extra training time does CrossNovo need compared to existing end-to-end methods?
>
>
> As mentioned in our response to Question 1, CrossNovo requires approximately 30% more training time compared to the baseline AT model. However, the additional computational cost during testing is minimal, with less than a 4% increase compared to existing AT models. Thank you again for this question!
>
> Let us know if there is any other questions, we would love to provide more information! Thank you again for your time!

---

> > ### Comment · Reviewer_irZV · 2024-11-23
> >
> > Thank you for your response.
> >
> > > We will include a discussion on the additional computational costs for both training and inference in our revised manuscript.
> > > We have noted these typos and will correct them in our revised manuscript.
> >
> > Given the 2 week review period, it would have been more convincing to directly revise the manuscript rather than promising future revisions.
> >
> > > Consequently, the time increase during inference, once the model is fully trained, is almost negligible.
> >
> > It would be helpful if this statement were supported by concrete evidence. Providing actual inference times for CrossNovo compared to other baselines would make the argument more convincing.
> >
> > > Our method demonstrates a significant improvement of approximately 3% in whole-sequence-level accuracy, as shown in benchmark progression. This improvement represents a substantial advance, pushing accuracy to new heights. Such a performance gain is noteworthy in this domain, where even incremental improvements have a major impact.
> >
> > The claim that 3% is a  "significant improvement" is unsupported and unconvincing, given that prior baselines have shown larger gains (e.g., Casa.V2 to Contra.).

---

### Official Review · Reviewer_dDE5 · 2024-11-01

**Soundness:** 3
**Presentation:** 2
**Contribution:** 3
**Rating:** 5
**Confidence:** 3

**Summary:**

This work introduces CrossNovo, a novel method for de novo peptide sequencing that integrates the advantages of both autoregressive (AT) and non-autoregressive (NAT) transformer architectures. The authors developed a joint training and distillation-based training approach that employs a shared encoder for both AT and NAT decoders. They also designed a novel cross-decoder attention mechanism that facilitates the transfer of knowledge from the NAT component to the AT component. The experiment results demonstrated that CrossNovo achieves superior performance on widely-used benchmarks for de novo peptide sequencing, surpassing existing AT and NAT models across various species and evaluation criteria.

**Strengths:**

The paper presents a novel hybrid approach combining AT and NAT models for peptide sequencing, featuring innovative cross-decoder attention and importance annealing techniques. It also provides comprehensive experiments and ablation studies. The clear structure and presentation enhance its accessibility. This work advances de novo peptide sequencing, with potential applications in other domains and implications for biological and medical research. The improved accuracy in peptide sequencing demonstrates the practical value of this contribution to proteomics.

**Weaknesses:**

There are a couple of points that can strengthen the paper.

- While the results are strong, the improvements over baselines are relatively modest in some cases (e.g. 1-3% on some species). The authors could provide more discussion on the practical significance of these improvements.
- The computational difficulties of CrossNovo compared to baselines is not discussed. Given the additional components, it would be worth clarifying any additional computational costs required in the training.
- The experiments focus solely on protein tasks. Exploring the applicability of the proposed approach to other general domain sequence generation tasks could broaden the impact.
- While the paper provides problem formulations, I think it’s still quite difficult to understand the problem backgrounds due to limited explanations contained in the paper. For example, the authors didn’t fully explain about the inputs set such as mass-to-charge ratios. For the completeness of the paper, it would greatly helpful if the authors can provide more problem backgrounds and explanations as the supplementary materials.

**Questions:**

1.	Can you provide insights into why the cross-decoder attention mechanism is effective? What kind of information is being transferred from NAT to AT?
2.	Given that NAT models struggle with varying sequence lengths, how does CrossNovo handle this issue? Can you provide any experiment results regarding this point?
3.	Have you explored using the proposed approach for other sequence generation tasks beyond peptide sequencing? If not, do you think it would be effective, and what modifications might be needed?

---

> ### Author Response · Authors · 2024-11-18
> **Reply to Reviewer dDE5**
>
> We first thank the reviewer for recognizing  our contribution and providing valuable feedbacks!
>
> Here are our reply to your questions:
>
>
> > While the results are strong, the improvements over baselines are relatively modest in some cases (e.g., 1–3% on some species). The authors could provide more discussion on the practical significance of these improvements.
>
> Thank you for this valuable comment. We appreciate the opportunity to further elaborate on the practical significance of our results. While the improvements may appear modest at first glance (e.g., 1–3% on certain species), we would like to emphasize their importance, both here and in our revised manuscript:
>
>
>
> + The Challenge of Peptide Sequencing Accuracy:
>
>   In peptide sequencing, an entire peptide sequence must match the target exactly to be considered correct. This contrasts with other sequence prediction tasks, such as NLP, where predictions do not need to match word-for-word to be considered acceptable. Even a single incorrect amino acid prediction renders the entire peptide sequence invalid, potentially leading to a completely different protein function. This "all-or-nothing" correctness requirement makes improvements in accuracy inherently more challenging.
>
>   As a result, every 1% improvement in peptide accuracy represents a significant number of correctly predicted peptide sequences, especially when applied to large sequencing datasets. For example, a 3% improvement indicates a substantial leap in the number of correctly sequenced peptides, reducing downstream errors in protein-level studies.
>
> + Historical Context and Difficulty of Improvement:
>
>   Previous de novo sequencing algorithms have been published in high-impact venues, including Nature Methods, Nature Machine Intelligence, Nature Communications, ICML, and AAAI. However, progress in peptide-level accuracy has historically been slow. Most prior works have reported only incremental improvements of 2–5%. For instance, the transition from LSTM-based models to Transformer architectures—a significant architectural change—yielded a modest improvement of roughly 4%.
>
>   This slow progression highlights the inherent difficulty of achieving substantial accuracy gains in this task. Nevertheless, even incremental progress, such as our observed 4% average improvement across species, represents a meaningful step forward. Over time, such step-by-step advances can accumulate to achieve high accuracy, bringing us closer to reducing sequencing errors and enhancing protein sequencing techniques for real-world applications.
>
>
>   Thank you again for raising this important point. We will expand on these discussions in our manuscript to better contextualize the significance of our improvements.
>
>
> > The computational difficulties of CrossNovo compared to baselines is not discussed. Given the additional components, it would be worth clarifying any additional computational costs required in the training.
>
> Thank you for highlighting the importance of discussing computational difficulties. We agree that this is a valuable suggestion, particularly for those aiming to replicate our results. The training process for our model involves two stages, and the additional computational time is a notable cost. Specifically, during joint training, we observed that both decoders converged faster than when trained separately. Training the AT decoder alone typically requires 130 epochs, with 1 million data samples per epoch, while the NAT decoder on its own requires 150 epochs. In joint training, the total number of epochs is reduced to 120, although each epoch takes approximately 1.3 times longer compared to training a single decoder. The second stage, which involves knowledge distillation, requires an additional 20 epochs. Overall, the total training time increases by approximately 30% when using the same setup of 8 A100 GPUs, compared to training a single AT model.
>
> In addition to time, the computational cost also increases with respect to GPU memory usage. With both decoders needing to be accommodated during joint training (without employing CPU offloading), the memory requirement is approximately 40% higher than for a single AT model. We will expand on these computational considerations in our manuscript to provide greater clarity. Thank you again for this constructive suggestion, which we believe will be highly beneficial for future researchers working on this task.

---

> > ### Author Response · Authors · 2024-11-18
> > **Reply**
> >
> > > The experiments focus solely on protein tasks. Exploring the applicability of the proposed approach to other general domain sequence generation tasks could broaden the impact.
> >
> >
> > Thank you for this insightful suggestion. While we believe our approach introduces a general methodology for combining bi-directional information with uni-directional models, which could potentially benefit a variety of sequence generation tasks, our research group is currently focused on protein-related research and operates within limited resources. Expanding our scope to domains such as language or image processing is beyond our current capacity. However, we agree that this represents an interesting direction for future research and could be explored by researchers specializing in other domains. Our approach provides a flexible and generalizable method that may serve as a foundation for broader applications.
> >
> >
> > That said, we would like to emphasize the importance of the specific task we are addressing: peptide sequencing. This task is fundamental to proteomics research, serving as the starting point for numerous downstream applications. Previous de novo sequencing algorithms have been featured in high-impact venues such as Nature Methods, Nature Machine Intelligence, Nature Communications, ICML, and AAAI, underscoring the significance and competitiveness of this field. We believe that the task we are tackling has substantial scientific value on its own, and we hope that our dedicated efforts will advance the field and contribute to impactful developments, similar to domain-specific breakthroughs like AlphaFold.
> >
> >
> > We sincerely thank the reviewer for this suggestion and will incorporate these points into our revised manuscript.
> >
> > >  While the paper provides problem formulations, I think it’s still quite difficult to understand the problem backgrounds due to limited explanations contained in the paper. For example, the authors didn’t fully explain about the inputs set such as mass-to-charge ratios. For the completeness of the paper, it would greatly helpful if the authors can provide more problem backgrounds and explanations as the supplementary materials.
> >
> > Thank you for this valuable feedback! We appreciate the suggestion to provide more detailed background information, as we’ve noticed that several reviewers have expressed a desire for greater clarity on this task.
> >
> >
> > Peptide sequencing is one of the primary methods for obtaining protein sequences from biological samples. For example, when a blood sample is submitted to a hospital or research lab, protein sequences are typically derived through mass spectrometry. This process involves analyzing the resulting spectra using algorithms like de novo sequencing to reconstruct peptide sequences.
> >
> >
> > To elaborate further, mass spectrometry works by measuring the mass-to-charge ratio (m/z) of ionized molecules. This ratio is a critical parameter that facilitates the identification of peptides and proteins. The m/z value is defined as the mass of an ion divided by its electrical charge. When a protein sample is digested into smaller peptide fragments, these fragments are ionized and introduced into the mass spectrometer. Inside the instrument, the ionized peptides are accelerated through an electric field, and their trajectories are influenced by their mass and charge.
> >
> >
> > By detecting how the peptides travel through the spectrometer, the device generates a spectrum that displays peaks corresponding to different m/z values. These peaks represent the peptide fragments in the sample. The intensities of the peaks provide additional information about the abundance of the fragments. By analyzing the patterns of these peaks and their intensities, algorithms can reconstruct the original peptide sequences.
> >
> >
> > This process is a cornerstone of protein sequencing efforts because the m/z values provide direct information about the composition and structure of the peptide fragments. The precision and detail of mass spectrometry make it an indispensable tool in proteomics research.
> >
> >
> > We will further expand on this explanation in the Appendix to ensure that readers have a comprehensive understanding of the task. We hope this additional background information helps to address your concerns, and we thank you again for bringing this to our attention!

---

> > > ### Author Response · Authors · 2024-11-18
> > > **Reply**
> > >
> > > > Given that NAT models struggle with varying sequence lengths, how does CrossNovo handle this issue? Can you provide any experiment results regarding this point?
> > >
> > > Thank you for raising this important question. NAT models indeed predict all positions simultaneously, requiring a predefined sequence length for generation. In contrast, CrossNovo, as an AT-based model augmented with features learned from NAT, retains the capability to adapt to varying sequence lengths, similar to any other AT model.
> > >
> > >
> > > Specifically, CrossNovo generates sequences in a next-token prediction manner. During generation, the model produces one token at a time and continues generating until it predicts an end-of-sequence token (<eos>). This token signals the model to stop, representing the end of the sequence. As a result, CrossNovo can dynamically generate sequences of varying lengths, accommodating longer sequences as needed, unlike NAT models that are constrained by predefined lengths.
> > >
> > >
> > > In terms of experimental results, it is important to note that in peptide sequencing tasks, the sequence length is generally limited to a maximum of 40 amino acids due to biological constraints, with an average sequence length of approximately 20. In our testing data, fewer than five sequences exceeded a length of 40, making long sequences exceptionally rare. For these outlier cases, CrossNovo successfully generated longer sequences, a task that NAT models were unable to accomplish due to their fixed-length constraint.
> > >
> > >
> > > However, we acknowledge that long sequences are exceedingly rare in this task, and many studies exclude them for simplicity. We will add a discussion about this in our manuscript to clarify how CrossNovo handles varying sequence lengths and highlight the model's ability to address these edge cases. Thank you again for pointing this out.
> > >
> > >
> > > > Have you explored using the proposed approach for other sequence generation tasks beyond peptide sequencing? If not, do you think it would be effective, and what modifications might be needed?
> > >
> > > Thank you for this thoughtful question. As discussed in response above, we have not explored applying our approach to other tasks. This is primarily due to our limited time and resources, as well as our research group’s focus on protein research and the significant importance of the sequencing task we are addressing.
> > >
> > > That said, we believe our method is general enough to be applicable to other sequence generation tasks. The approach was not designed specifically for protein sequences but instead leverages bi-directional information, which has been shown to be valuable in many domains, including language tasks that are inherently directional.
> > >
> > > Applying our method to other domains would likely require minimal modifications. The primary changes would involve adapting the tokenization and embedding strategies to suit the specific characteristics of the new domain. For instance, our current model uses protein-specific tokenization and embedding tailored to peptide sequences. For other tasks, such as language modeling or image-based sequence generation, domain-specific adaptations for these components would be necessary.
> > > We believe exploring these adaptations would be an interesting avenue for future research, and we hope our method provides a foundation for such investigations.
> > >
> > > Thank you again for raising this important point! We hope these reply align with the reviewer’s expectations and a higher score would be appreciated if reviewer thinks our points are valuable! We welcome reviewer for further discussion and questions as well! Thank you again for your time!

---

> > > > ### Comment · Reviewer_dDE5 · 2024-11-26
> > > > **Post-Rebuttal**
> > > >
> > > > I appreciate the authors' effort on the response and clarifications. On my side, I don't have any more comments to be addressed.
> > > >
> > > > Unfortunately, however, after reading the other reviewers' comments, I also generally agree with their concerns on the lack of novelty and limited scope. Thus, I decided to lower my score to 5.

---

### Official Review · Reviewer_j6dc · 2024-11-03

**Soundness:** 2
**Presentation:** 2
**Contribution:** 2
**Rating:** 3
**Confidence:** 4

**Summary:**

The paper proposes a two-stage approach for mass-spectrum based De Novo peptide sequencing where the sequence of a particular peptide is predicted given its mass spectrum as input. The authors seem to combine the advantages of both autoregressive transformer (AT) and non-autoregressive transformer (NAT) for this task, by using the more efficient AT for decoding while incorporating the bidirectional representations from NAT. The first stage involves jointly training an autoregressive decoder an a non-autoregressive decoder to output the peptide’s amino acid sequence prediction. The second stage of supervised training involves concatenating the output representation of the non-autoregressive to the mass spectrum representations for the AT to condition on them during its decoding. Experiments on two benchmark datasets show that the proposed approach outperforms baseline approaches on most tasks.

**Strengths:**

Approach shows better empirical performance than baseline approaches

Combining the strength of both non-autoregressive transformer and auto-regressive transformer is an interesting idea

**Weaknesses:**

Limited Novelty: It seems to the reviewer that the approach is combining the representations from a trained non-autoregressive transformer (NAT) with the mass spectrum representations for the autoregressive transformer (AT) to condition on. It does not seem that there is distillation involved as what the distillation term is generally understood in the ML community. Rather, the proposed approach is a two-stage supervised training where the second stage is a finetuning process by adding the NAT representations as additional input representation for the AT decoder.

Application-specific studies: While combining the strengths of NAT and AT have the potential to impact various applications, only mass spectrum based de novo peptide sequencing is studied here which might only be of interest to a small subset of the ML community in this venue.

Clarity: Rationale and explanation of the proposed approach can be further improved (see questions below). The paper is at times challenging to follow even though the concept is actually rather simple.

Questionable design choice: the choice of CTC loss is rather uncommon for peptide and protein modeling but not explained or backed by experiments.

**Questions:**

Why is CTC loss selected over cross-entropy for NAT training even though cross-entropy is generally used for protein modeling? Why is only CTC used for NAT but not for AT training? It is common for peptides and proteins to have consecutive similar amino acid where they should not be merged; why would this be appropriate to merge these amino acid here? The reviewer would suggest an ablation study to support this choice.

Why is equation 1 used to encode the mass spectrum? How is v there related to mass-to-charge ratio and intensities, mz_i and g_i?

What is h_t^{update}? Does it refer to the AT decoder’s representation at the cross decoder attention block?
Typo: Line 249: docoders -> decoders

---

> ### Author Response · Authors · 2024-11-18
> **Reply to Reviewer j6dc**
>
> > It does not seem that there is distillation involved as what the distillation term is generally understood in the ML community. Rather, the proposed approach is a two-stage supervised training where the second stage is a finetuning process by adding the NAT representations as additional input representation for the AT decoder.
>
> Thank you for your thoughtful comment regarding the use of the term "Knowledge Distillation" (KD). We categorized our work under the broader scope of KD because, in a general sense, we understand KD as leveraging a better-performing "teacher" model to guide a "student" model in improving task performance. In our case, this involves utilizing the stronger performance of the trained Non-Autoregressive Transformer (NAT) model in certain species to enhance the performance of the Autoregressive Transformer (AT) model (the "student").
>
>
> We recognize, however, that our use of the term KD may deviate from the strict definition commonly accepted in the machine learning community. Traditionally, KD involves using the soft labels or logits produced by the teacher model as supervision for the student, rather than leveraging intermediate latent features. We acknowledge that our interpretation of KD might be viewed as an extended or unconventional usage. If this terminology is contentious, we are more than willing to adjust it in our manuscript to align with community standards. Thank you for pointing this out!!
>
> In addition to addressing KD, our work introduces several novel ideas specifically targeting this task, to address your question on novelty:
>
> + The Cross-Decoder Attention Module was designed to capitalize on prior findings that NAT and AT excel in different scenarios due to architectural inductive biases. Unlike  approaches that combine methods arbitrarily (e.g., A+B), our design is deeply rooted in biological insights and leverages the complementary strengths of NAT and AT across different species and data distributions.
>
> + Many aspects of our model, such as the encoding of spectra and peptides, shared encoders, and joint training, are informed by biological insights and tailored to improve sequencing performance. For example, joint training was implemented based on our observation that AT and NAT encourage distinct encoding features for input spectra, allowing their strengths to complement each other effectively.
>
>
> We believe the significance of our work lies not only in the novel design elements but also in its practicality and usability. Specifically, our model delivers impactful results that directly benefit the research community by providing a tool that can be readily applied to real-world biological research (open sourced in provided link)
>
> > Application-specific studies: While combining the strengths of NAT and AT have the potential to impact various applications, only mass spectrum based de novo peptide sequencing is studied here which might only be of interest to a small subset of the ML community in this venue.
>
> We thank the reviewer for raising concerns about the domain coverage of our work. To clarify, we would like to provide some context about the scope and significance of the de novo peptide sequencing task.
>
> Peptide sequencing is currently one of the primary methods for obtaining protein sequences from biological samples. For example, when a blood sample is submitted to a hospital or research lab, protein sequences are almost always derived through mass spectrometry. This process involves analyzing the resulting spectra using algorithms like de novosequencing to extract peptide sequences. Such methods are widely used in numerous labs and medical institutions, playing a critical role in proteomics research.
>
> De novo peptide sequencing serves as the starting point for downstream analyses, such as protein structure prediction (e.g., AlphaFold), making it a fundamental task in the field. Over the past 2–3 years, de novo sequencing algorithms have been featured in high-impact venues such as [Nature Methods](https://www.nature.com/articles/s41592-018-0260-3), [Nature Machine Intelligence](https://www.nature.com/articles/s42256-023-00738-x), [Nature Communications](https://www.nature.com/articles/s41467-024-49731-x), [ICML](https://icml.cc/virtual/2022/spotlight/16796), and [AAAI](https://ojs.aaai.org/index.php/AAAI/article/view/27765),
> These publications underscore the competitive nature and importance of advancing performance in this area.
>
> Our work builds on this well-established task, setting a new state-of-the-art performance compared to all prior methods, as verified against published baselines. Furthermore, we provide an open-source implementation of our code and model, making our tool accessible for a wide range of proteomics-related research applications.
>
> We hope this additional information on our task can clarify the importance of this task and the domain we work on!

---

> > ### Author Response · Authors · 2024-11-18
> > **Reply**
> >
> > > Clarity: Rationale and explanation of the proposed approach can be further improved (see questions below).
> >
> > > Why is CTC loss selected over cross-entropy for NAT training even though cross-entropy is generally used for protein modeling?
> >
> > Thank you for this insightful question. The choice of CTC (Connectionist Temporal Classification) loss over cross-entropy (CE) for training Non-Autoregressive Transformers (NAT) stems from the fundamental difference in how NAT and Autoregressive Transformers (AT) model probabilities. CTC loss has been a long-standing convention in the NAT community and has dominated in fields like speech recognition and machine translation over the past 10–20 years. Below, we provide a detailed explanation of why CTC is essential for NAT training.
> >
> > In NAT models, generation is non-conditional, meaning the prediction of the ${i+1}^{th}$ token is not strictly dependent on the i$^th$ token. This independence allows NAT models to predict all tokens in parallel, unlike AT models, which predict tokens sequentially, with p($y_{i+1}$∣$y_{1:i}$). Consequently, NAT models estimate p(y$_i$) directly. This non-conditional generation introduces a well-known issue referred to as the multi-modality problem.
> >
> > To illustrate, let us consider machine translation as an example, where CTC is a standard loss for NAT models:
> >
> > Suppose we aim to translate the French sentence "au revoir" into English. In the training corpus, there might be two acceptable translations: "good bye" and "see you".
> >
> > In AT models, once the first token (e.g., "see") is predicted, the subsequent token probabilities are conditioned on it (e.g., the second token will have higher probabilities for "you"). This ensures that "see you" is decoded naturally due to the sequential conditioning.
> >
> > In contrast, NAT models generate token probabilities independently and simultaneously for each position. For example:
> >
> > At position one, the probability distribution might be: {see: 0.6, good: 0.3, other words: 0.1}.
> >
> > At position two, the distribution might be: {you: 0.4, bye: 0.5, other words: 0.1}.
> >
> > If argmax decoding is used, the resulting translation could be "see bye", which is semantically incoherent. This arises because NAT lacks token-level connections, leading to the multi-modality problem where independent token predictions mix distributions from different valid sequences.
> >
> > Cross-entropy loss exacerbates this issue in NAT because it optimizes tokens independently. For example:
> > For position one, both "good" and "see" receive positive optimization signals during training as long as they match the training corpus, without regard for subsequent positions.
> >
> > Similarly, for position two, generating "you" or "bye" reduces the loss, irrespective of what was generated at position one.
> > This independent optimization ignores the global connections between tokens, resulting in incoherent outputs like "see bye".
> >
> > CTC loss mitigates this issue by focusing on sequence-level optimization. Unlike cross-entropy, CTC does not assign rewards for correctly predicting a token in isolation. Instead, the model must generate the entire correct sequence to minimize the loss.
> > For example, if the target sequence is "goodbye", the NAT model must output "goodbye" or a reducible form such as "good _ bye" to reduce the CTC loss. Generating sequences like "see _ bye" will not reduce the loss, as the sequence-level match fails.
> > This sequence-level emphasis ensures that NAT models capture global connections across tokens, compensating for the lack of conditional dependencies inherent in NAT architectures. CTC is therefore well-suited for NAT, as it enforces global coherence in output sequences, unlike cross-entropy, which focuses on token-level accuracy.
> >
> > We hope this explanation clarifies why CTC loss is the preferred choice for training NAT models in our work.

---

> > > ### Author Response · Authors · 2024-11-18
> > > **Re**
> > >
> > > > Why is only CTC used for NAT but not for AT training? It is common for peptides and proteins to have consecutive similar amino acid where they should not be merged; why would this be appropriate to merge these amino acid here?
> > >
> > > This distinction arises from two main reasons:
> > >
> > > + Conditional Probability in AT Models:
> > >   As mentioned earlier, AT models inherently use conditional probabilities, where the prediction of each token depends on the tokens generated before it. Once a previous token is fixed, the probability distribution for the next token adjusts accordingly. This sequential dependency naturally enforces token-to-token connections in AT models, eliminating the need for additional mechanisms, such as CTC loss, to enhance global connections between tokens.
> > >
> > > + Static vs. Dynamic Token Probabilities:
> > >
> > >   The calculation of CTC loss requires static token probabilities, which are independent of prior token choices. This is necessary to compute all possible generational paths (e.g., "Good Good Bye," "Good Bye Bye," etc.), which can reduce to "Good Bye" under the CTC merging rule. NAT models generate token probabilities independently for each position, making them static and suitable for CTC loss calculation.
> > >
> > >   In contrast, AT models generate token probabilities dynamically, where the probability of the next token is conditioned on the previously generated tokens. For example, to calculate the CTC path probability of "Good Bye Bye," the AT model would first decode "Good Bye," and only then calculate the probability of the third token. This dynamic nature of token probabilities would require performing inference as many times as there are possible CTC paths, which is computationally infeasible.
> > >
> > >   Therefore, CTC loss is specifically designed for NAT models, as they produce static probability distributions at each position and benefit from enhanced inter-token connections provided by CTC.
> > >
> > > > Why consecutive similar amino acids should not be merged in peptide sequencing?
> > >
> > > It is common for peptides and proteins to contain consecutive similar amino acids, which must be represented distinctly without being merged. In one of our tested datasets, consecutive repeated tokens (e.g., 2-grams like "LL") accounted for 24% of all 2-grams in the sequences.
> > >
> > > This differs from natural language, where large token vocabularies (e.g., 50,000 tokens for LLaMA) result in less frequent token repetition. Amino acids, on the other hand, have a small vocabulary of just 20 tokens. To encode different protein functions, amino acids are often repeated consecutively in sequences (e.g., the peptide "ALLPT," where "L" appears twice).
> > >
> > > To address this, CTC loss incorporates a placeholder token, ϵ, to avoid merging consecutive similar amino acids. For instance, in the peptide "ALLPT," a valid CTC path would be "ALϵLPT," ensuring that the two "L" tokens remain distinct.
> > > CTC also allows for some error tolerance by permitting certain paths to merge tokens, such as "AALϵLPT," where "A" is repeated but later merged. This flexibility accommodates the uncertainty in token positioning that can arise during unconditional generation. By merging redundant tokens, the model reduces errors while maintaining the accuracy needed for sequence-level optimization. This design has been highly effective in reducing errors in NAT generation.
> > >
> > > We sincerely thank the reviewer for their detailed and technical questions. We hope our explanations clarify your concerns regarding the use of CTC loss and its application in peptide sequencing. Please feel free to reach out with any further questions or feedback.

---

> > > > ### Author Response · Authors · 2024-11-18
> > > > **Re**
> > > >
> > > > > Why is equation 1 used to encode the mass spectrum? How is v there related to mass-to-charge ratio and intensities, mz_i and g_i?:
> > > >
> > > > Equation 1 uses a sinusoidal function to encode floating-point numbers into a dd-dimensional vector. This approach is analogous to the positional encoding used in transformer models to convert integer positions into vectors. The choice of this specific encoding function was driven by the following considerations:
> > > >
> > > > + The mz values in the spectrum peaks can vary greatly, ranging from as small as 1–10 to as large as 2000–3000. Normalizing mz values into a fixed range (e.g., 0–1) would distort the absolute differences between values, even if relative magnitudes were preserved. However, absolute differences are critical for determining chemical substances in the spectrum. Therefore, it is essential to encode the original mz values directly without applying normalization.
> > > >
> > > >
> > > >   A naive approach, such as mapping mz values to a high-dimensional space using a linear layer (e.g., mapping from a float to a 512-dimensional vector), would disproportionately favor larger mz values. This occurs because matrix multiplication inherently assigns greater weight to larger numbers, leading smaller values to have diminished influence in subsequent neural network computations.
> > > >
> > > > + The sinusoidal function in Equation 1 offers a robust workaround. It maps each floating-point number to a dd-dimensional vector, where each dimension is represented by sin⁡sin and cos⁡cos values within the range [−1,1]. This ensures that all input values, regardless of magnitude, are mapped into a uniform distribution, preventing larger values from overshadowing smaller ones in matrix multiplications.
> > > >
> > > >   Additionally, sinusoidal encoding preserves the relative magnitude between numbers. This is because sin⁡sin and cos⁡cos functions oscillate between −1 and 1 at specific frequencies for each dimension, allowing relative differences to be captured through variations in oscillation patterns. For example, in the first dimension, a frequency of π might encode even numbers as −1 and odd numbers as 1, while other dimensions oscillate at different frequencies to reflect finer-grained differences. This property makes sinusoidal encoding particularly effective for capturing relationships between mz values.
> > > >
> > > > #### Clarifying the Role of $v_i$:
> > > >
> > > > In Equation 1, v  is a placeholder for the floating-point numbers being encoded, which in this case are the mz​ (mass-to-charge ratios) and g (intensities). To avoid redundancy in the equation, v  was used to represent these values generically. However, we recognize that this notation might cause confusion. We will revise the equation to explicitly use mz and g​ to improve clarity.
> > > > We apologize for not explaining these choices clearly in our original write-up.
> > > >
> > > > Thank you for pointing this out, and we will ensure that these reasons are thoroughly addressed in our revised manuscript.
> > > >
> > > > > 3.4 What is h_t^{update}? Does it refer to the AT decoder’s representation at the cross decoder attention block? Typo: Line 249: docoders -> decoders
> > > >
> > > >
> > > > Yes, your understanding is correct. h^update represents the updated features after integrating the NAT decoder's outputs into the AT decoder's features, as shown in Equation 6. This update incorporates cross-attention to both the encoder representation and the NAT representation within the AT model.
> > > >
> > > >
> > > > We appreciate the reviewer’s careful reading and for pointing out the typo on line 249 ("docoders" -> "decoders"). We will correct this in the revised manuscript. Thank you for bringing it to our attention!

---

> ### Author Response · Authors · 2024-11-18
> **Re**
>
> > Questionable design choice: the choice of CTC loss is rather uncommon for peptide and protein modeling but not explained or backed by experiments:
>
> Thank you for raising this concern. The use of CTC loss in this task follows the precedent set by previous state-of-the-art [model](https://www.biorxiv.org/content/10.1101/2024.05.17.594647v1), as briefly mentioned in our paper. The effectiveness of CTC loss for protein modeling has been [extensively studied](https://www.biorxiv.org/content/10.1101/2024.05.17.594647v1) in earlier works on this task. We will expand on this discussion in our manuscript to provide further clarity and context.
>
>
> To summarize the rationale for using CTC loss (also detailed in our response to Question 3), CTC enables global sequence-level optimization rather than focusing solely on token-level accuracy. This is critical for peptide sequencing, as the task demands the entire amino acid sequence, including the precise order of residues, to be correct. Any error in the amino acid sequence can result in functional and property changes in the peptide. CTC aligns well with this requirement by prioritizing sequence correctness, ensuring that the model captures global dependencies and relationships.
>
>
> Moreover, CTC loss has been demonstrated to perform exceptionally well in this task by previous state-of-the-art models, and we have adopted it as part of our pipeline to build on their success.
>
>
> We thank the reviewer once again for their thorough evaluation of our work and for catching these detailed points. We hope our response clarifies the rationale behind this design choice and provides a better understanding of our contributions. If our responses address the reviewer’s concerns, we would greatly appreciate a higher score if the reviewer finds our points valuable. We are happy to engage in further discussions or address any additional questions. Thank you once again for your time and thoughtful review!

---

> ### Comment · Reviewer_j6dc · 2024-11-26
> **Rebuttal**
>
> Thank you for the response and clarification. My main concern of lack of novelty and limited scope remains so I will keep my score.

---

### Official Review · Reviewer_9efx · 2024-11-03

**Soundness:** 3
**Presentation:** 2
**Contribution:** 2
**Rating:** 3
**Confidence:** 3

**Summary:**

Note: I am not familiar with proteins research. However, I have given my best analysis of the work from a methodical perspective.

CrossNovo is a framework to autoregressive (AT) de novo peptide sequencing by distilling knowledge from non-autoregressive (NAT) models. This allows the model to capture bidirectional protein representation for decoding. Leveraging a shared encoder and cross-decoder attention, the approach combines strengths of both model types, purportedly achieving superior performance across multiple benchmarks. The study aims to address NAT's limitations with sequence generalization and optimization by combining AT’s sequential generation with NAT's bidirectional capabilities. Experimental results show state-of-the-art performance on two datasets, demonstrating improved amino acid precision and peptide recall. CrossNovo is suggested as a valuable tool for proteomics research.

**Strengths:**

* Combines AT with NAT in a novel way
* Achieves State-of-the-art performance
* Comprehensive benchmarking
* Detailed experiments

**Weaknesses:**

This reads like an engineering paper. It largely focuses on ensemble and engineering strategies without introducing a novel methodology. The use of a shared encoder and cross-decoder attention mechanism, while useful, does not represent a fundamental advance in model architecture for peptide sequencing.

Moreover, I am concerned that such engineering project could have led to extensive evaluations and finetuning, raising concerns about potential overfitting.

There might be ways to combine NAT and AT, but I believe the proposed approach is quite naive and has no theoretical grounding. It might be a result of the very short sequences that it works, I would like to see biological evaluations on longer sequences.

In general, for this conference I'd expect evaluations across diverse biological benchmarks with longer sequences to highlight how deep learning extends to proteomics.

**Questions:**

I'd like to see a comparison with the size of models for previous work, I want to know that this is not just a result of a larger model capacity.

---

> ### Author Response · Authors · 2024-11-18
> **Reply to Reviewer 9efx**
>
> > I am not familiar with proteins research. However, I have given my best analysis of the work from a methodical perspective.
>
> We sincerely thank the reviewer for their time and effort in reading our manuscript and providing valuable feedback. We appreciate your comments regarding your background and your insights into protein-related tasks. To provide additional context (background) and highlight the significance of our work, we would like to briefly summarize the task of de novo peptide sequencing. Thank you in advance for your time in reviewing this explanation.
>
>
> Peptide sequencing is currently the primary methods for obtaining protein sequences from biological samples. For example, when a blood sample is submitted to a hospital or research lab, the protein sequences are typically derived through mass spectrometry. This involves analyzing the resulting spectra using algorithms like de novo sequencing to extract the sequences.
>
>
> Given its role as the starting point for downstream analyses, such as protein structure prediction (e.g., AlphaFold), the task of peptide sequencing is fundamental in proteomics research. Previous de novo sequencing algorithms have been published in high-impact venues such as [Nature Methods](https://www.nature.com/articles/s41592-018-0260-3), [Nature Machine Intelligence](https://www.nature.com/articles/s42256-023-00738-x), [Nature Communications](https://www.nature.com/articles/s41467-024-49731-x), [ICML](https://icml.cc/virtual/2022/spotlight/16796), and [AAAI](https://ojs.aaai.org/index.php/AAAI/article/view/27765), showcasing significant advancements and highly competitive performance in this field.
>
>
> Our work builds on same task problem, setting a new state-of-the-art performance benchmark compared to all prior work mentioned above (as verified against published baselines). Additionally, we provide an open-source implementation of our code and model weights (linked in paper), making our tool accessible for a wide range of proteomics-related research applications. We hope this context helps emphasize the importance of our contributions. Thank you once again for your thoughtful review.
>
> Below, we address all reviewer’s concern point-by-point:
>
> >  This reads like an engineering paper. It largely focuses on ensemble and engineering strategies without introducing a novel methodology. The use of a shared encoder and cross-decoder attention mechanism, while useful, does not represent a fundamental advance in model architecture for peptide sequencing.
>
> We appreciate the reviewer’s question regarding the novelty and engineering aspects of our work, we’d like to further clarify this as follows:
>
> - For protein-based biological research, usability and performance are paramount considerations, often surpassing other factors. Tools developed for this field are heavily utilized by biologists directly in lab analyses and protein discovery workflows. For instance, the baseline tool DeepNovo [Nature Methods](https://www.nature.com/articles/s41592-018-0260-3) has seen extensive application in such tasks, since developed.
>
>   Our work demonstrates a significant improvement of approximately 3% in whole-sequence-level accuracy, as evident from benchmark progression. This represents a substantial advance, pushing accuracy to a new heights. Such a performance gain is noteworthy and immediately applicable to biological research through our open-sourced software.
>
>   It is also important to note that engineering in this domain—unlike traditional NLP or CV tasks—requires extensive trial and error due to the specific challenges of bioinformatics. We have dedicated significant effort to ensuring the system works robustly, which we believe is a valuable contribution in itself.
>
> - other two points of this question continues in next thread, thanks for reading!

---

> > ### Author Response · Authors · 2024-11-18
> > **Cont.**
> >
> > Continues previous points :
> >
> > - Our work introduces several novel ideas specifically targeting this task: Cross-Decoder Attention module is designed to exploit prior findings that Non-Autoregressive Transformers (NAT) and Autoregressive Transformers (AT) achieve high performance in different scenarios due to architectural inductive biases. Unlike many engineering approaches that arbitrarily combine methods (A+B), our design is rooted in biological insights and findings, aiming to leverage the strengths of both NAT and AT in this task for different species and data distributions.
> >
> >   Many of our other model components, such as the encoding of spectra and peptides, shared encoders, and joint training, are grounded in biological insights and optimized for the specific goal of improving sequencing performance. For example, joint training was implemented because we observed that AT and NAT encourage distinct encoding features on input spectra. We acknowledge that our paper may not have made these design motivations as explicit as they should be in writing, and we will revise the manuscript to provide clearer explanations of why each part of the model was adopted. To the best of our knowledge, this is the first work to combine AT and NAT in this way, we believe we have offered novelty in both bio field and AI research.
> >
> > - If one examines prior works in this domain published in venues such as [ICML](https://icml.cc/virtual/2022/spotlight/16796),  [AAAI](https://ojs.aaai.org/index.php/AAAI/article/view/27765), and N[Nature Methods](https://www.nature.com/articles/s41592-018-0260-3), most approaches rely on very clear and simple models (e.g., standard Transformers or LSTMs). In contrast, our work presents a significantly more advanced model design, achieving much higher performance as well.
> >
> >   We believe the value of the work lies not only in fancy design but also in its practicality and usability. Specifically, our model delivers impactful results that benefit the community by providing a tool that can be directly applied to real-world research. We believe this aligns with the goals of other applied fields, such as medical imaging.
> > It would be unfortunate if our work were evaluated solely on the perceived “fanciness” of its methodology. We hope the reviewer will reconsider our contributions in light of the progress achieved and the impact on the field. We thank the reviewer in advance for reading this long comment regarding the first question!
> >
> > > Moreover, I am concerned that such engineering project could have led to extensive evaluations and finetuning, raising concerns about potential overfitting.
> >
> > We appreciate the reviewer’s concern regarding potential overfitting. Below, we provide a detailed explanation to address this point:
> >
> > Our algorithm was developed in strict adherence to the training data protocols established in prior work, ensuring that there is no overlap between the peptides in the training dataset and those in the testing datasets. Specifically, we evaluated our model on four distinct testing datasets from various sources: 9Species v1, v2, Antibody data for Humans, and Antibody for mice (shown in the 4 tables in our paper). The 9Species datasets, in particular, encompass data from nine different species with diverse distributions. All datasets were carefully curated to guarantee zero overlap with the training data, ensuring that the testing phase accurately reflects the model's generalization performance rather than any memorization of training data.
> >
> > Also, the use of multiple datasets with varying distributions mitigates the risk of overfitting to a specific dataset or species. The consistent 3–5% performance gain observed across these diverse datasets highlights the generalizability of our model to various types of peptide data. This indicates that our model is robust and not limited to a narrow distribution of data.
> >
> > Reviewers and other researchers are also welcome to download released model weight to even test on any of their own data, as we have released the model weights and code in the provided link, enabling independent validation of our results on any new peptide data.
> >
> > We believe that the strict separation of data, combined with the consistent performance gains across diverse testing datasets, demonstrates the true capability of our model and effectively addresses concerns about overfitting. Thank you for raising this important point, and we welcome any additional feedback or suggestions on this!

---

> ### Author Response · Authors · 2024-11-18
> **Reply to reviewer**
>
> > It might be a result of the very short sequences that it works, I would like to see biological evaluations on longer sequences.
>
> We apologize for not providing sufficient context regarding the characteristics of peptide sequencing tasks. To clarify, a peptide is a short protein sequence, and the reason peptides are sequenced instead of entire proteins lies in the mechanism of mass spectrometry. This process involves enzymatically breaking down proteins into charged fragments, effectively splitting long protein chains into very short pieces.
>
> In peptide sequencing, the length of a peptide almost never exceeds 30, with only a small fraction reaching lengths of 35–40. Once the peptides are sequenced, they are assembled back into proteins using separate algorithms, which is a different task entirely. This means that peptide sequencing as a task does not involve handling long sequences.
>
> Consequently, algorithm design for peptide sequencing focuses specifically on shorter sequences, with an average length of around 20. This approach has been standard across all existing algorithms for this task. Long-sequence modeling, on the other hand, is typically employed in different tasks such as protein modelling (using LLM) and de novo protein design. The biological nature of peptides inherently restricts the task to most sequences of length 10–20 in both training and testing datasets.
>
>
> Although the sequences are short, peptide sequencing presents a unique challenge: every single token must be predicted correctly. Even a single incorrectly predicted amino acid can lead to errors in protein assembly. This stringent accuracy requirement makes the task highly challenging despite not involving long sequences.
> We hope this context provides additional clarity and helps the reviewer better appreciate the value of our work. Thank you for the opportunity to address this important point.
>
>
> > I'd expect evaluations across diverse biological benchmarks with longer sequences to highlight how deep learning extends to proteomics.
>
> As mentioned in our response to the previous question, the entire field of peptide sequencing is inherently focused on peptides with lengths of up to approximately 40 amino acids. This limit is reflected in the datasets we have evaluated, which include peptides of varying lengths from 10 to 40 amino acids, with an average length of around 20.
>
> Peptides longer than this threshold are rare because they tend to be enzymatically shortened during the mass spectrometry process. This is due to the inherent instability of longer peptide chains, which are fragmented into shorter, charged pieces to facilitate sequencing.
> We apologize for not clarifying this background information earlier and will ensure that these details are more explicitly explained in our revised manuscript. Thank you again for raising this question—it has helped us identify an opportunity to make our work more transparent and accessible.
>
> > I'd like to see a comparison with the size of models for previous work, I want to know that this is not just a result of a larger model capacity.
>
> We thank the reviewer for raising the question regarding the model size. For this task, we have used a model with the exact same size as the baseline Transformer model. Specifically, our final Autoregressive Transformer (AT) network consists of 33 million parameters, as the Non-Autoregressive Transformer (NAT) decoder serves only as an auxiliary component. If the NAT decoder is included, the total model size amounts to 45 million parameters.
>
>
> Our architecture comprises 9 Transformer layers for both decoders, with a hidden size of **400** dimensions. This is consistent with the baseline **NAT** model, PrimeNovo (33 million parameters and same number of layers and dimensions). In comparison, the previous best-performing **AT** models in our baseline—ContraNovo and Casanovo V2—are larger, with model sizes of 42 million parameters (**512** dimensions, 9 layers, with additional contrastive module) and 38 million parameters, respectively. Notably, our AT model is smaller in size due to the reduced dimension, making it more efficient while maintaining state-of-the-art performance.
>
> We also confirm that all training data used in our experiments are identical to those used in prior work. The full details of our model configurations, along with the codebase, are publicly available for further verification.
>
> We hope this response sufficiently addresses your concerns and provides clarity regarding the novelty and engineering contributions of our work. Thank you again for your thoughtful review and feedback. We look forward to any additional questions or discussions. We sincerely hope that the additional information provided allows you to reassess our work and offer a revised evaluation on previous scores. Thank you once again for your time and consideration.

---

> > ### Comment · Reviewer_9efx · 2024-11-21
> > **Re: Reply to Reviewer.**
> >
> > I have read the other reviews. In particular, it seems that reviewer  j6dc has similar concerns to mine. Below are follow-up questions.
> >
> > "For protein-based biological research, usability and performance are paramount considerations, often surpassing other factors."
> >
> > Not at ICLR, maybe at NAR or bioinformatics. For ICLR I expect methodological advances, not web servers.
> >
> > "Our work demonstrates a significant improvement of approximately 3% in whole-sequence-level accuracy"
> >
> > Which is impressive, I would leave it to the AC on whether this is adequate for ICLR. Personally, I would like to see novel methodological development at this conference, not engineering projects and webservers, they are better served at bio-oriented journals.
> >
> > "For example, joint training was implemented because we observed that AT and NAT encourage distinct encoding features on input spectra. "
> >
> > This seems interesting, I would be more inclined to accept this paper if you had a more detailed analysis of why combining the models the way you do make sense. If not, it seems like a kitchen sink method. I.e. you just combined them without much reasoning.
> >
> > "To the best of our knowledge, this is the first work to combine AT and NAT in this way, we believe we have offered novelty in both bio field and AI research."
> >
> > Any method can be combined, this does not make it particularly interesting.
> > Moreover, if you are the first one proposing a new methodological combination of AT and NAT, as reviewer j6dc mentioned, why only consider this narrow dataset? There are plenty of public datasets of interests to the wider ICLR audience.

---

> > > ### Author Response · Authors · 2024-11-22
> > > **Reply to Reviewer**
> > >
> > > We sincerely thank the reviewer for their further feedback and for raising detailed questions and concerns regarding our novelty.
> > > We understand the reviewer’s apprehension that our approach might appear to combine methods in a rudimentary way without deriving insights. In response, we provide a detailed explanation to clarify that our design is rooted in meaningful insights, not a superficial combination of methods. We appreciate your time and consideration in reviewing our clarification.
> > >
> > >
> > > In the context of protein (peptide) sequences, species exhibit vastly different protein distributions. For example, humans have specific amino acid arrangements that appear more frequently compared to bacteria. This variation underscores the importance of evaluating models across diverse protein datasets from different species. Performance on a single distribution, such as human data, does not necessarily guarantee good performance on other species, such as mouse data.
> > >
> > >
> > > During our initial experiments, we observed that autoregressive transformer (AT) models consistently outperformed non-autoregressive transformer (NAT) models on human data, a critical species given its extensive use in downstream tasks. As shown in Table 1, at both the amino acid and peptide levels, the ContraNovo (AT) model achieves a significant performance advantage (approximately 3%) over the PrimeNovo (NAT) model for human data.
> > >
> > > Conversely, NAT models performed significantly better on data from other species. Repeated experiments with varied seeds and hyperparameters consistently revealed this performance gap, indicating that the differences stem from inductive biases inherent to the model architectures.
> > >
> > >
> > > While NAT models currently represent state-of-the-art performance in this task, their relative weakness on human data hinders their broader applicability. Motivated by this observation, we designed our new model aiming to combine the strengths of both architectures, as highlighted in our introduction and experiments.
> > >
> > >
> > > Our experimental results validate this approach. Our model successfully integrates the advantages of both AT and NAT architectures across all species. Specifically, in species where NAT models traditionally excel, our distilled AT model now matches or surpasses their performance. Meanwhile, on human data—which is particularly critical—our model retains the AT model’s superior performance, even achieving slight improvements.
> > >
> > >
> > > To further investigate, we conducted an analysis of the subsets of test data solved by each model. Among 10,000 samples tested per species, we found a 10% difference in the data subsets correctly solved by AT and NAT models, despite their overall accuracy differing by only 3%.
> > >
> > > This suggests that each model specializes in distinct data subsets. By employing our distillation strategy, we successfully transfer the NAT model’s inductive biases to the AT model. Our analysis shows that the AT model, post-distillation, can now correctly solve many of the test cases previously solvable only by the NAT model. This demonstrates that our approach is not a blind combination of two architectures but a deliberate integration designed to harness the strengths of both.
> > >
> > >
> > > We hope this detailed explanation addresses your concerns and clarifies the rationale behind our approach. Please let us know if there are any additional questions or points we can further elaborate on. Thank you once again for your valuable feedback!

---

> > > > ### Comment · Reviewer_9efx · 2024-11-23
> > > > **Re: rebuttal**
> > > >
> > > > I see, thank you for the clarification.
> > > >
> > > > It seems to me that the model is a type of ensemble between heterogeneous models. I.e. because their design and training are vastly different, they most likely will find different solution spaces and have high entropy between predictions as opposed to just training multiple seeds of the same model. Moreover, ensembling autoregressive predictors is non-trivial, and this poses a strategy to do such, which I think is quite interesting and definitely of interest to the broader research community.
> > > >
> > > > However, I do not think the current paper, combination approach, and even motivation (i.e. their validation performance differs) are the level of methodological rigor I want to accept this paper. I.e. interesting research direction, but lack of execution.
> > > >
> > > > My score stays the same.

---

### Meta-Review · Area_Chair_vThW · 2024-12-18

**Metareview:**

While Autoregressive Transformers excel in language generation tasks, Non-Autoregressive Transformers outperform them in certain biology tasks like protein modeling due to their bidirectional information flow but face challenges with generalization and scalability. To bridge this gap, the paper proposes a framework for distilling NAT knowledge into ATs using joint training with a shared encoder, a specialized cross-decoder attention module, and a novel training pipeline with important annealing and gradient blocking. The proposed method combines the strengths of ATs and NATs, achieving good performance across all metrics in de novo peptide sequencing.

Most reviewers acknowledge the writing and comprehensive experimental validation. However, the major concerns consistently lie in the limited innovation and scope. We recommend a reject.

**Additional Comments On Reviewer Discussion:**

During the rebuttal period, the authors did a good job and put in lots of effort. We believe incorporating all the discussions into the revision could be very beneficial.

---

### Decision · Program_Chairs · 2025-01-22

Reject